# Hepatocytes differentiate into intestinal epithelial cells through a hybrid epithelial/mesenchymal cell state in culture

Shizuka Miura[1], Kenichi Horisawa[1], Tokuko Iwamori[1], Satoshi Tsujino[1], Kazuya Inoue[1], Satsuki Karasawa[1], Junpei Yamamoto[1], Yasuyuki Ohkawa[2], Sayaka Sekiya[1] & Atsushi Suzuki[1] ✉

Hepatocytes play important roles in the liver, but in culture, they immediately lose function and dedifferentiate into progenitor-like cells. Although this unique feature is well-known, the dynamics and mechanisms of hepatocyte dedifferentiation and the differentiation potential of dedifferentiated hepatocytes (dediHeps) require further investigation. Here, we employ a culture system specifically established for hepatic progenitor cells to study hepatocyte dedifferentiation. We found that hepatocytes dedifferentiate with a hybrid epithelial/mesenchymal phenotype, which is required for the induction and maintenance of dediHeps, and exhibit Vimentin-dependent propagation, upon inhibition of the Hippo signaling pathway. The dediHeps re-differentiate into mature hepatocytes by forming aggregates, enabling reconstitution of hepatic tissues in vivo. Moreover, dediHeps have an unexpected differentiation potential into intestinal epithelial cells that can form organoids in three-dimensional culture and reconstitute colonic epithelia after transplantation. This remarkable plasticity will be useful in the study and treatment of intestinal metaplasia and related diseases in the liver.

Hepatocytes constitute nearly 80% of the liver mass and have a wide range of functions, including metabolism, formation and secretion of bile, drug detoxification, glycogen storage, and protein synthesis. A unique characteristic of primary hepatocytes is that they rapidly lose their functional properties in monolayer culture[1,2] and exhibit immaturity as a result of dedifferentiation into progenitor-like cells through the epithelial-mesenchymal transition (EMT)[3,4].

Recent lineage-tracing analyses have revealed that adult human and mouse hepatocytes undergo dedifferentiation and give rise to biliary progenitor cells in vivo by responding to severe chronic liver injury[5–7]. The Hippo signaling pathway is a fundamental regulator of in vivo hepatocyte dedifferentiation. Activation of the Hippo transducer yes-associated protein (Yap) in hepatocytes is sufficient to induce hepatocyte dedifferentiation through activation of Notch signaling[8], which activates the biliary program in adult hepatocytes[5,6,9].

It was recently reported that hepatocyte dedifferentiation in monolayer culture results from mechanical tension-induced Yap activation[10] and is supported by activation of canonical Wnt signaling, inhibition of transforming growth factor β (TGF-β) signaling, and hypoxia[11–14].

Induction of hepatocyte dedifferentiation will be a possible therapeutic strategy for the treatment of liver diseases. However, studies examining the cellular dynamics of hepatocyte dedifferentiation, mechanisms underlying the induction and maintenance of a dedifferentiated state, and differentiation potential of hepatocyte-derived dedifferentiated cells are still required.

In the developing liver, hepatic progenitor cells (also known as hepatoblasts) proliferate to give rise to both hepatocytes and cholangiocytes as descendants to form liver tissues[15]. Meanwhile, adult liver progenitor cells (i.e., oval cells) appear in response to excessive liver impairment and are able to propagate and differentiate into both

[1]Division of Organogenesis and Regeneration, Medical Institute of Bioregulation, Kyushu University, Fukuoka 812-8582, Japan. [2]Division of Transcriptomics, Medical Institute of Bioregulation, Kyushu University, Fukuoka 812-8582, Japan. ✉e-mail: suzukics@bioreg.kyushu-u.ac.jp

hepatocytes and cholangiocytes to restore damaged liver tissue[16]. We previously isolated hepatoblasts and oval cells from developing and chronically injured mouse livers, respectively, by combining flow cytometry and fluorescence-conjugated antibodies[17–19]. In our culture system, hepatoblasts and oval cells identified by fluorescence-assisted cell sorting were cultured in individual wells of 96-well plates, and clonal colonies formed from each cell were expanded in a long-term culture. These data suggest that the culture condition enabling efficient expansion of hepatoblasts and oval cells is suitable for the culture of hepatic progenitor cells. In this study, we used this culture system established for hepatic progenitor cells to culture adult mouse hepatocytes and examine the dynamics and mechanisms of hepatocyte dedifferentiation and the differentiation potential of cells dedifferentiated from hepatocytes.

Our present data demonstrate that hepatocytes isolated from adult mouse livers dedifferentiate into expandable progenitor-like cells under culture conditions used for hepatic progenitor cells. These dedifferentiated hepatocytes, termed dediHeps, exhibit a hybrid epithelial/mesenchymal (E/M) phenotype in monolayer culture, which is dependent on the inhibition of Hippo signaling, but not on the activation of TGF-β signaling, and is required for acquisition and maintenance of their dedifferentiated state. Although dediHeps stably propagate with immature phenotypes in monolayer culture, they stop proliferating and re-differentiate into mature hepatocytes by forming aggregates in three-dimensional (3D) culture and can reconstitute hepatic tissues after transplantation into injured livers. Moreover, we found that dediHeps have remarkable plasticity; dediHeps can give rise to intestinal epithelial cells, which can form intestinal organoids under

3D culture conditions and reconstitute colonic epithelial tissues after transplantation into a chemically-induced colonic injury model.

## Results

### Expansion of hepatocyte-derived cells in culture

We crossed mice expressing an inducible form of Cre recombinase (*CreER*[T2]) from the *albumin* (*Alb*) genomic locus [*Alb-CreER*[T2] mice[20]] with *R26R*[YFP/YFP] reporter mice[21]. In the double-mutant mice, administration of tamoxifen (TM) permanently marked *Alb*-positive hepatocytes and enabled tracing of their fates.

To observe hepatocytes in culture, we isolated them from the livers of TM-treated *Alb-CreER*[T2];*R26R*[YFP/+] mice and cultured them by serial passaging. Three hours after their initial plating, co-immunofluorescence analyses revealed that all YFP-positive cells expressed both Alb and Hnf4α, but YFP was not detected with the cholangiocyte marker cytokeratin 19 (CK19; also known as Krt19) (Fig. 1a, b). Thus, we have developed an experimental system for tracing the fate of hepatocytes in culture that involves highly specific isolation of hepatocytes and their heritable labeling with YFP.

In monolayer culture, the morphology of some hepatocytes shifted from round to spindle-shaped within 4 days after plating; these cells continued to proliferate and gradually became small epithelial cells after several weeks of culture (Fig. 1c). These YFP-positive hepatocyte-derived cells could be maintained through serial passaging (Fig. 1c). Two independent experiments demonstrated that YFP was expressed in more than 99% of cultured cells derived from the liver of TM-treated *Alb-CreER*[T2];*R26R*[YFP/+] mice, indicating that their hepatocyte progenies can proliferate and expand in long-term culture

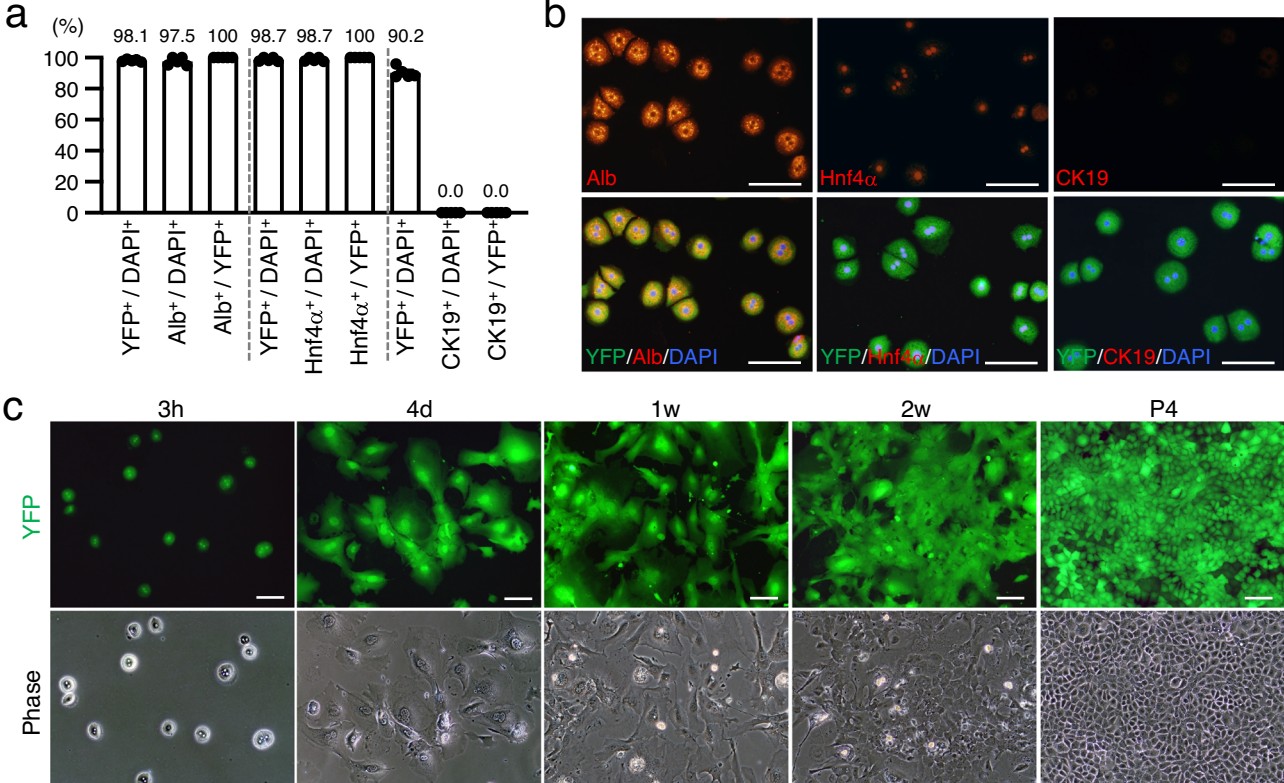

**Fig. 1 | Propagation of hepatocyte-derived cells in monolayer culture.**
**a** Hepatocytes isolated from adult mouse livers were stained with antibodies against YFP and Alb, Hnf4α, or CK19, 3 h after plating. DNA was stained with DAPI. The graph shows the percentages of YFP⁺, Alb⁺, Hnf4α⁺, and CK19⁺ cells in the DAPI⁺ population and the percentages of Alb⁺, Hnf4α⁺, and CK19⁺ cells in the YFP⁺ population. Data represent the mean ± SEM (*n* = 5 independent experiments).

Source data are provided as a Source Data file. **b** Representative micrographs of isolated hepatocytes expressing YFP and Alb, Hnf4α, or CK19. DNA was stained with DAPI. **c** Representative fluorescence micrographs and morphologies of YFP⁺ hepatocytes 3 h, 4 days (d), 1 week (w), and 2 w after plating and at passage 4 (P4). Scale bars, 100 μm.

(Supplementary Fig. 1a, b). Moreover, similar to hepatocytes isolated form *Alb-CreER^T2;R26R^YFP/+* mice, those from untreated wild-type mice were also capable of propagating in culture (Supplementary Fig. 1c). Thus, these cells were named dedifferentiated hepatocytes (dediHeps).

## dediHeps have a hybrid E/M phenotype

During conversion of primary hepatocytes to dediHeps, the expression of Alb and the epithelial cell marker E-cadherin (E-cad; also known as Cdh1) was maintained (Fig. 2a). Along with Alb and E-cad, CK19 and the mesenchymal cell marker Vimentin (Vim) were expressed in spindle-shaped hepatocyte-derived cells 4 days after plating; their expression was maintained in long-term dediHep cultures (Fig. 2a). Confocal microscopy clearly demonstrated that Alb and E-cad were co-expressed with CK19 and Vim, respectively, in dediHeps (Fig. 2b). These findings suggest that primary hepatocytes adopt a hybrid E/M phenotype during the early stages of monolayer culture, with this phenotype maintained during and after the transition from hepatocyte to dediHep. Expression levels of *Cdh1* and the epithelial cell markers *claudin-3 (Cldn3)* and *occludin (Ocln)* were not changed, but expression of *Vim* and the mesenchymal cell markers *S100 calcium binding protein A7A (S100a7a)* and *laminin B1 (Lamb1)* increased, in cultured dediHeps relative to expression in hepatocytes freshly isolated from adult mouse livers (Fig. 2c). In addition, expression of the proliferation marker Ki67 (also known as *MKi67*) and the cholangiocyte/progenitor marker *prominin-1 (Prom1*; also known as *CD133*) was increased and expression of several genes encoding liver enzymes was decreased in dediHeps relative to their expression in hepatocytes, while both dediHeps and hepatocytes expressed similar levels of transcription factors involved in hepatocyte differentiation, including *Hnf4α* and *Hnf1α* (Fig. 2c). Kyoto Encyclopedia of Genes and Genomes (KEGG) pathway enrichment analyses revealed that genes associated with the cell cycle and metabolic pathways were enriched among those that were up- and down-regulated, respectively (Fig. 2d). Sequential gene expression analyses during hepatocyte-to-dediHep conversion revealed that the expression of genes encoding liver enzymes began to decrease after 1 day in monolayer culture, followed by subsequent increases in the expression of *MKi67, Krt19, Prom1*, and mesenchymal cell markers (Fig. 2e). Taken together, our data demonstrate that dediHeps maintain a subset of key characteristics of hepatocytes and epithelial cells, but lose function and acquire a partial mesenchymal phenotype.

## dediHeps re-differentiate into hepatocytes and reconstitute liver tissues after transplantation into injured livers

To investigate re-differentiation potential, we developed dediHep aggregate cultures, as aggregation has been shown to facilitate the maturation of immature hepatocyte-like cells derived from pluripotent stem cells[22] or directly induced from fibroblasts[23]. Under 3D culture conditions, dediHeps gradually assembled to form uniform aggregates (Fig. 3a), expressing Alb and E-cad, but neither Vim nor Ki67 (Fig. 3b). To assess the hepatic maturation of dediHep aggregates, we performed gene expression analyses in them, with monolayer controls, and conduced KEGG pathway enrichment analyses. The expression of genes encoding liver enzymes was increased, but expression of *MKi67, Krt19, Prom1*, and mesenchymal cell markers was decreased, in aggregates (Fig. 3c). The changes in gene expression were promptly induced starting from 1 day of 3D culture (Supplementary Fig. 2). Meanwhile, the expression levels of *Cdh1, Cldn3, Ocln, Hnf4α*, and *Hnf1α* were almost unchanged between aggregates and monolayers (Fig. 3c). KEGG pathway enrichment analyses revealed that genes differentially expressed in dediHep aggregates versus monolayers were significantly enriched for those associated with metabolic pathways (up-regulated) and cell cycle (down-regulated) (Fig. 3d). The hepatic maturation of dediHep aggregates could also be induced from dediHeps propagated in long-term monolayer culture (Supplementary Fig. 3).

In order to determine whether liver parenchyma could be reconstituted by dediHep-derived, re-differentiated hepatocytes, we intrasplenically injected dissociated aggregates from TM-treated *Alb-CreER^T2;R26R^YFP/+* mice into the livers of fumarylacetoacetate hydrolase (*Fah*)-deficient (*Fah^-/-*) recipient mice, which is a mouse model of hereditary tyrosinaemia type I[24]. *Fah^-/-* livers are severely damaged without the provision of 2-(2-nitro-4-trifluoromethylbenzoyl)−1,3-cyclohexanedione (NTBC); therefore, these mice can be used as recipients to analyze hepatocyte differentiation potential and hepatic tissue reconstitution. Three months after transplantation, dediHep-derived, YFP-positive cells reconstituted hepatic tissues of *Fah^-/-* recipients as Fah-positive hepatocytes and were morphologically indistinguishable from normal hepatocytes (Fig. 3e). These dediHep-derived hepatocytes did not express the mesenchymal cell markers Vim and α-smooth muscle actin (αSMA) (Supplementary Fig. 4). Tissue reconstitution capacity in dediHep-derived hepatocytes was similar to that of hepatocytes freshly isolated from adult mouse livers (Fig. 3f). These data demonstrate that aggregation eliminates the mesenchymal phenotype in dediHeps, and facilitating re-differentiation into growth-arrested, functional hepatocytes.

## dediHeps have an intestinal phenotype in monolayer culture

Given that dediHeps are undifferentiated cells with a hybrid E/M phenotype, we wondered whether they are able to differentiate into additional cell types. We discovered that more than 60% and 90% of dediHeps express the intestinal master transcription factor Cdx2 and its mRNA, respectively (Fig. 4a, b), normally expressed in intestinal epithelial cells but not in hepatocytes. Cdx2 was co-expressed with Alb in dediHeps (Fig. 4c) and gradually disappeared from dediHeps during re-differentiation into hepatocytes (Supplementary Fig. 5). In the development, bone morphogenetic protein (BMP) signaling is essential for mid/hindgut specification from definitive endoderm, inducing *Cdx2* expression and the fate of intestine[25,26]. Also, ectopic expression of the BMP target gene *inhibitor of differentiation (Id) 2* in the developing stomach induces *Cdx2*-positive intestinal epithelial cells in the gastric epithelia[27]. Thus, we examined whether BMP signaling is activated and critical for *Cdx2* expression, in dediHeps. Sequential gene expression analyses during hepatocyte-to-dediHep conversion revealed that expression of *BMP4, BMP7*, and the BMP target genes *Id1* and *Id2* was increased in dediHeps (Fig. 4d). Moreover, inhibition of BMP signaling by Noggin significantly suppressed the expression level of *Cdx2* in dediHeps (Fig. 4e). These data demonstrate that dediHeps express *Cdx2* through activation of BMP signaling in monolayer culture.

## dediHeps differentiate into intestinal epithelial cells that reconstitute colonic epithelia upon transplantation

The expression of Cdx2 in dediHeps suggested that they can differentiate into intestinal epithelial cells. To test this hypothesis, we embedded dediHeps in Matrigel 3D culture and treated them with epidermal growth factor (EGF), Noggin, and R-spondin1 (designated ENR) in the presence of Wnt3a and the Wnt agonist CHIR99021 (glycogen synthase kinase-3 inhibitor, designated WCENR), which induces fetal intestinal progenitor cells (FIPCs) to form spherical organoids (SOs)[28]. After 11 days of culture, dediHeps formed a small number of SOs, which could be propagated by serial passaging (Fig. 5a). The expression level of *Cdx2* was increased and those of *Alb* and *Vim* were decreased after the formation and passages of SOs (Fig. 5b). These SOs were composed of actively proliferating, polarized intestinal epithelial cells that express E-cad basolaterally, Villin (also known as Vil1) apically, and nuclear Cdx2 and Sox9 (Fig. 5c and Supplementary Fig. 6a), similar to FIPC-derived SOs[28,29]. Also, dediHep-derived SOs were formed by cells expressing CK19 and Hnf4α but not Alb, indicating that these SOs were not biliary organoids that express CK19 but not Hnf4α and Alb (Supplementary Fig. 6b). Global gene expression analyses revealed that genes

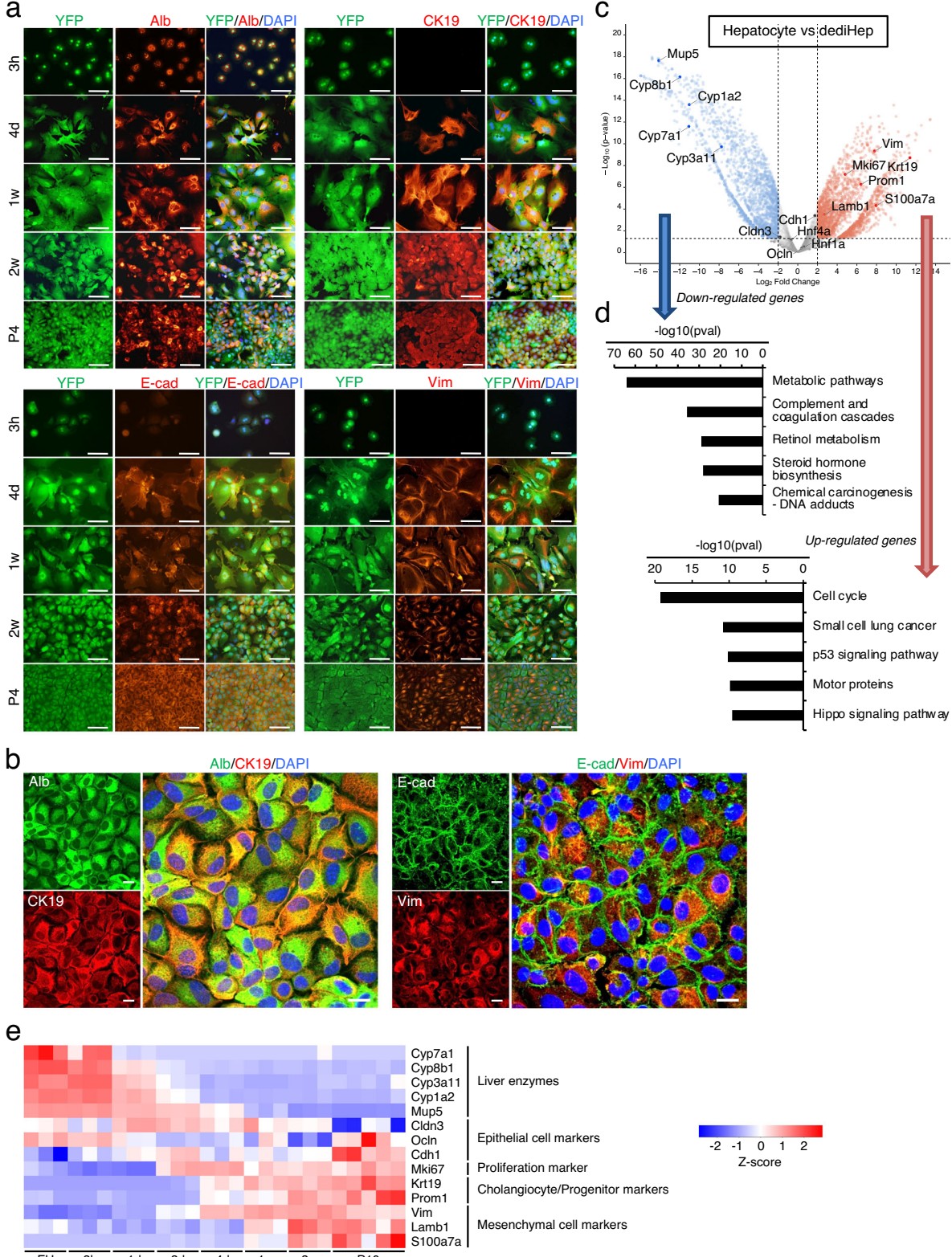

**Fig. 2 | dediHeps exhibit a hybrid E/M phenotype. a** Co-immunofluorescence staining (YFP with Alb, CK19, E-cad, or Vim) of hepatocytes in monolayer culture 3 h, 4 days (d), 1 week (w), and 2 w after plating and at passage 4 (P4). **b** Representative confocal microscopic micrographs of dediHeps that are immunoreactive for Alb with CK19 and for E-cad with Vim at P4. DNA was stained with DAPI. Scale bars, 100 μm (**a**) and 20 μm (**b**). **c** Volcano plot depicting differences in gene expression between dediHeps and hepatocytes. Each dot represents one gene. Blue and red dots represent down- and up-regulated genes, respectively, in dediHeps. Gray dots represent genes that are not significantly differentially expressed. **d** Top five most significantly enriched pathways associated with down- and up-regulated genes in dediHeps compared to hepatocytes, identified by KEGG pathway enrichment analysis. **e** Heatmap image from CEL-seq2 data showing the change in expression of the indicated genes during conversion of freshly isolated hepatocytes (FH) to dediHeps in monolayer culture 3 h, 1 d, 2 d, 4 d, 1 w, and 2 w after plating and at P10. Statistical difference was determined by quasi-likelihood F-test (**c**) or Fisher's exact test (**d**).

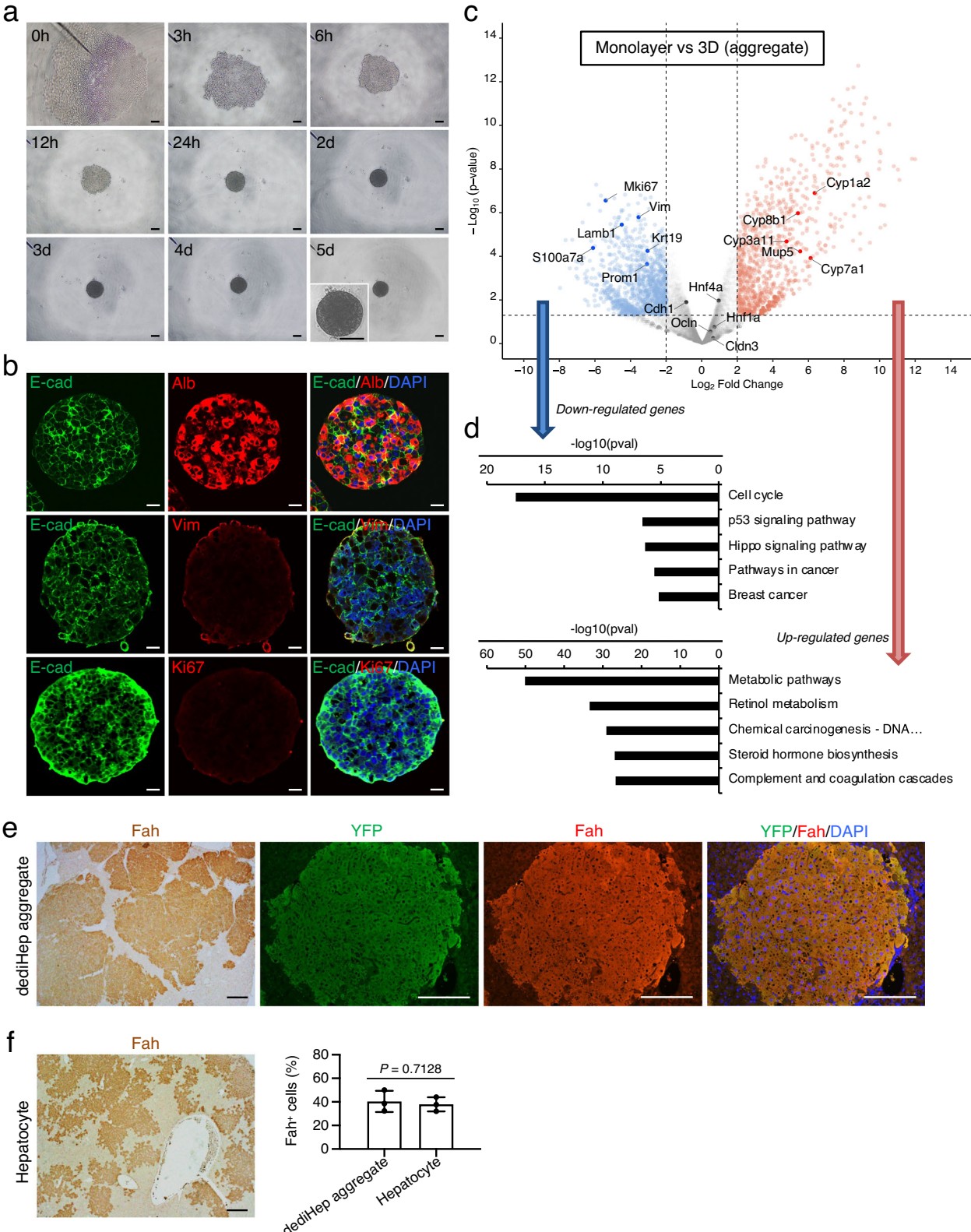

specifically expressed in dediHep-derived SOs or FIPC-derived SOs, compared with hepatocytes, are similar (Fig. 5d), and more than 97% of the detected genes were similarly expressed in both types of SOs (Fig. 5e). Moreover, addition of forskolin (FSK) with or without the cystic fibrosis transmembrane conductance regulator (CFTR) inhibitor CFTRinh-172 to the culture media blocked or induced the swelling of dediHep-derived SOs, respectively (Supplementary Fig. 7). Thus, epithelial cells forming dediHep-derived SOs can

functionally respond to FSK in a CFTR-dependent manner and increase the size of the SOs.

In addition to our in vitro analyses, we sought to assess whether dediHep-derived intestinal epithlialal cells could contribute to tissue reconstitution in vivo. It is known that FIPCs contribute to regeneration of adult colonic epithelium after transplantation into a chemically-induced colonic injury model[29]. Thus, we examined whether dediHep-derived intestinal epithelial cells behaved similarly. To this end,

**Fig. 3 | Re-differentiation of dediHeps into functional and transplantable hepatocytes in 3D culture. a** Representative morphology of a dediHep aggregate at the presented hours (h) and days (d) after initiation of 3D culture. Inset, the enlarged micrograph of aggregate. **b** Co-immunofluorescence staining (E-cad with Alb, Vim, or Ki67) of dediHep aggregates 5 days after initiation of 3D culture. **c** Volcano plot depicting differences in gene expression between dediHep aggregates and dediHep monolayer cultures. Each dot represents one gene. Blue and red dots represent down- and up-regulated genes, respectively, in dediHep aggregates. Gray dots represent genes that are not significantly differentially expressed. **d** Top five most significantly enriched pathways associated with down- or up-regulated genes in dediHep aggregates compared to dediHep monolayer cultures, identified by KEGG pathway enrichment analysis. **e** Immunohistochemical staining of Fah and co-immunofluorescence staining of YFP with Fah on liver sections of *Fah⁻/⁻* mice 3 months after transplantation of cells dissociated from dediHep aggregates derived from TM-treated *Alb-CreER^T2;R26R^YFP/+* mice. **f** A representative micrograph of Fah⁺ cells on liver sections of *Fah⁻/⁻* mice 3 months after transplantation of hepatocytes freshly isolated from the liver of adult wild-type mice. Graph at right depicts percentages of Fah⁺ cells in CK8/18⁺ epithelial cells on liver sections of *Fah⁻/⁻* mice 3 months after transplantation of cells dissociated from dediHep aggregates or hepatocytes. Data represent the mean ± SEM (*n* = 3 independent experiments). DNA was stained with DAPI. Scale bars, 200 μm (**a**, **e**, and **f**) and 20 μm (**b**). Statistical difference was determined by quasi-likelihood F-test (**c**), Fisher's exact test (**d**), or two-sided Student's *t* test (**f**). Source data are provided as a Source Data file.

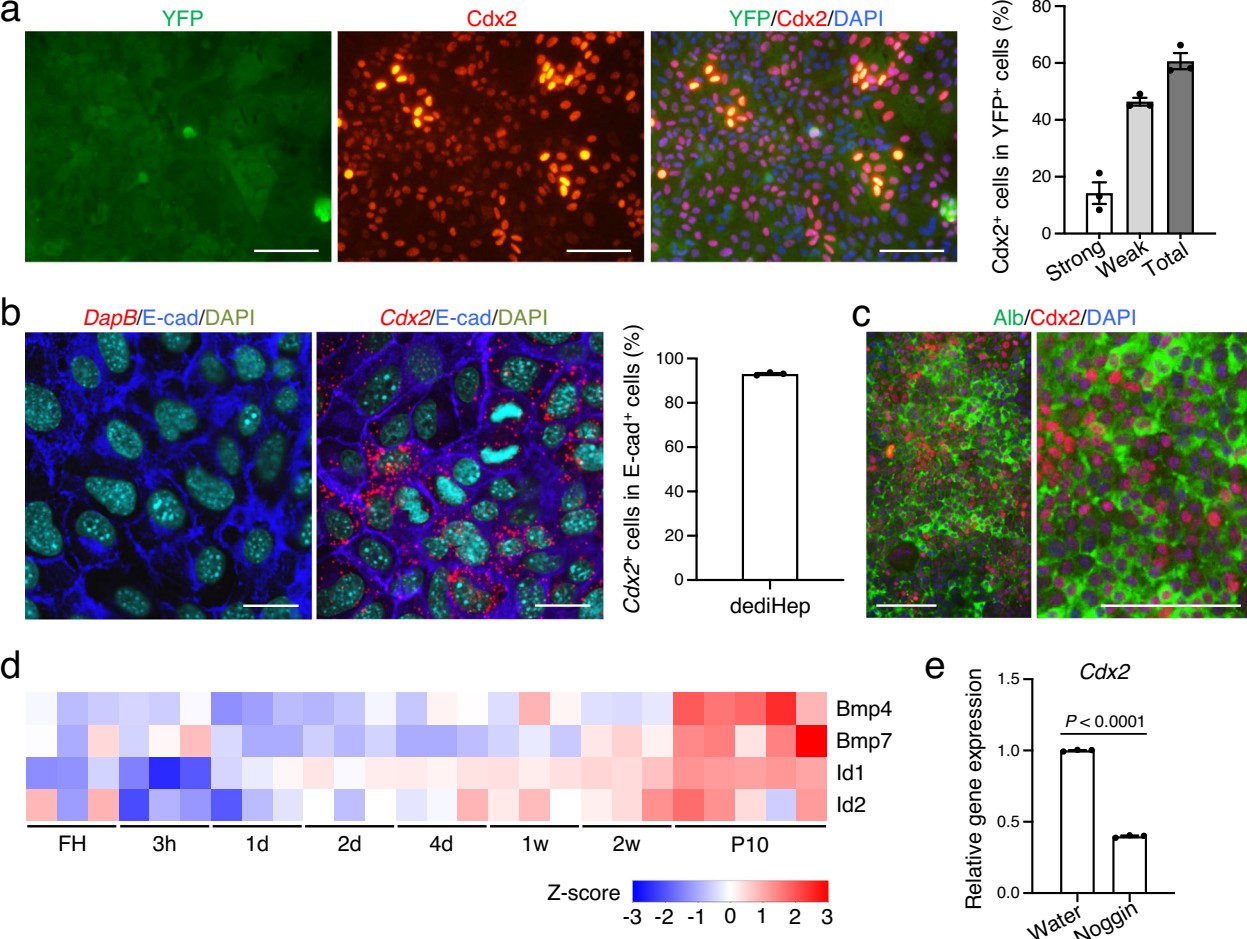

**Fig. 4 | dediHeps express *Cdx2* in response to BMP signaling. a** Co-immunofluorescence staining (YFP with Cdx2) of dediHeps in monolayer culture. Graph at right depicts percentages of cells that were strongly or weakly positive for Cdx2 in YFP⁺ cells. **b** *Cdx2* mRNA, but not the bacterial gene *DapB* mRNA (negative control), was detected in dediHeps by fluorescence in situ hybridization. E-cad protein was also detected in these dediHeps by immunofluorescence staining. Graph at right depicts percentages of *Cdx2*⁺ cells in E-cad⁺ cells. **c** Co-immunofluorescence staining (Alb with Cdx2) of dediHeps in monolayer culture. **d** Heatmap image from CEL-seq2 data showing the change in expression of the indicated genes during conversion of freshly isolated hepatocytes (FH) to dediHeps in monolayer culture 3 h, 1 day (d), 2 d, 4 d, 1 week (w), and 2 w after plating and at passage 10 (P10). **e** qPCR analysis of *Cdx2* expression in dediHeps cultured with water or the inhibitor of BMP signaling Noggin for 4 days in monolayer culture. All data were normalized to the values for dediHeps cultured with water and are depicted as fold-changes. Statistical difference was determined by two-sided Student's *t* test. DNA was stained with DAPI. Scale bars, 100 μm (**a**, **c**) and 20 μm (**b**). Data represent the mean ± SEM (*n* = 3 independent experiments). Source data are provided as a Source Data file.

dediHep-derived SOs were transplanted into the colons of immuno-deficient NOD/SCID/gamma (NSG) mice with dextran sulfate sodium (DSS)-induced acute colitis[29,30]. In mice transplanted with dediHep-derived SOs, donor cell clusters were macroscopically observed in colonic tissues 3 months after transplantation (Fig. 5f). Histology revealed that donor-derived cells were characterized as colonocytes expressing Cdx2 and Klf5 and gave rise to Villin⁺ absorptive cells, Muc2⁺ goblet cells, and EphB2⁺ and Sox9⁺ cells residing in the bottom of crypts (Fig. 5g). The body weights of mice transplanted with dediHep-derived SOs recovered faster than those of control mice (Fig. 5h). These data indicate that dediHep-derived SOs are capable of repopulating colonic epithelial tissues, similar to FIPC-derived SOs.

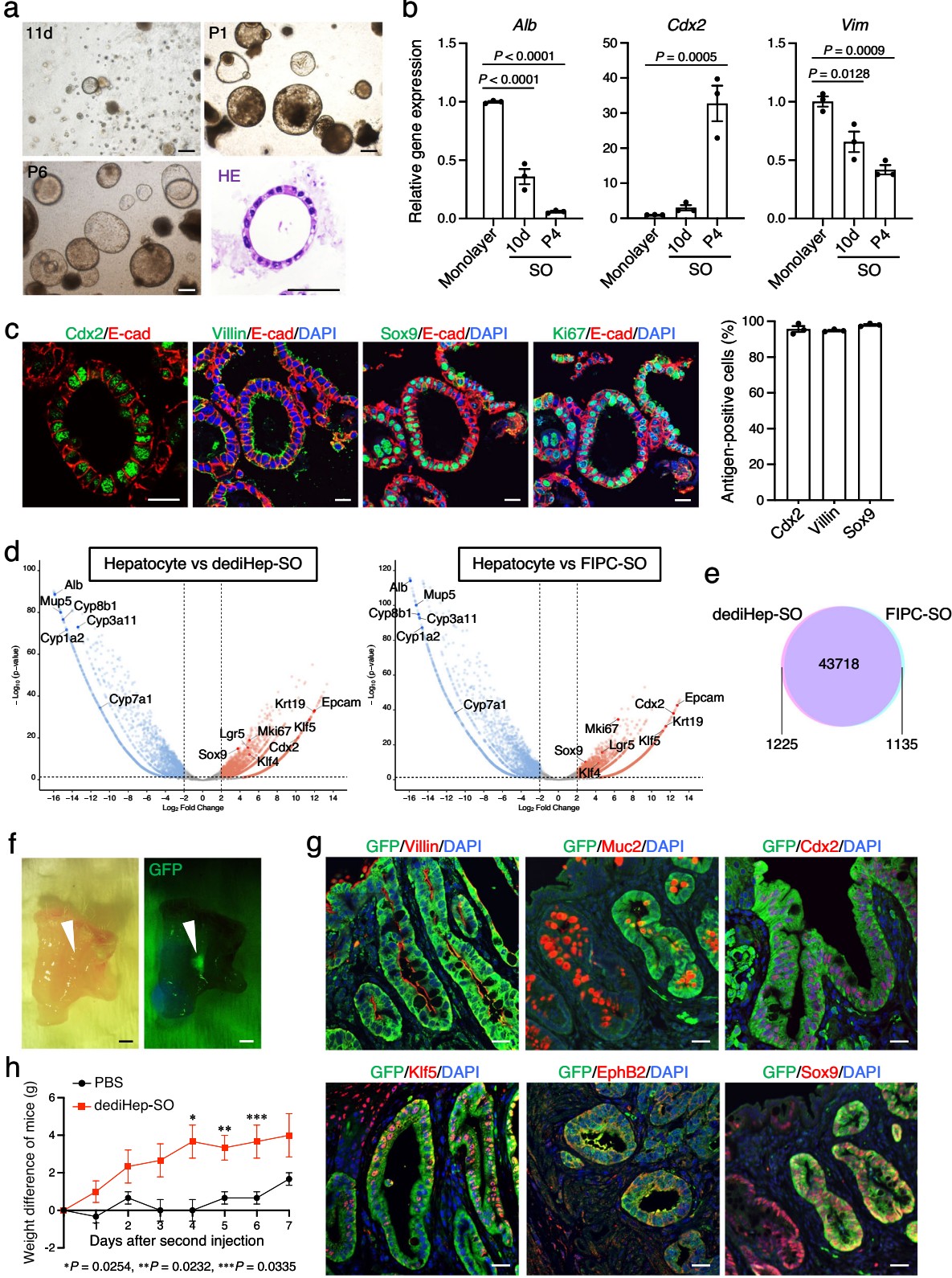

Taken together, our data demonstrate that dediHeps can differentiate into not only hepatocytes, but also intestinal epithelial cells.

## dediHeps give rise to intestinal stem cell (ISC)-like cells in response to forced expression of *Cdx2*

ISCs that reside at the bottoms of crypts in the small intestine continuously give rise to multilineage intestinal epithelial cells, including absorptive enterocytes, hormone-secreting enteroendocrine cells, mucin-secreting goblet cells, and antimicrobial protein-secreting Paneth cells, for replacing the epithelial cells lost by apoptosis at the villus tip[31]. ISCs form budding organoids (BOs) under 3D culture conditions, which consist of a lumen surrounded by a villus-like epithelial monolayer with budding crypt-like domains (CLDs)[32]. ISC-derived BOs are maintained in long-term culture by passaging with

**Fig. 5 | dediHeps give rise to intestinal epithelial cells capable of reconstituting colonic epithelia upon transplantation. a** Representative morphologies of dediHep-derived SOs at 11 days (d), passage 1 (P1), and P6 after initiation of 3D culture. Panel at lower-right is a representative micrograph of a hematoxylin and eosin (HE)-stained dediHep-derived SO. **b** qPCR analyses of expression of *Alb*, *Cdx2*, and *Vim* in dediHeps maintained in monolayer culture and SOs formed by dediHeps at 10 d and P4 after initiation of 3D culture. All data were normalized to the values for dediHeps in monolayer culture and are depicted as fold-changes. **c** Co-immunofluorescence staining (E-cad with Cdx2, Villin, Sox9, or Ki67) of dediHep-derived SOs. Graph at right depicts percentages of Cdx2⁺ cells, Villin⁺ cells, and Sox9⁺ cells in dediHep-derived SOs. **d** Volcano plots depicting differences in gene expression between hepatocytes and dediHep-derived SOs or FIPC-derived SOs. Each dot represents one gene. Blue and red dots represent down- and up-regulated genes, respectively, in dediHep-derived SOs or FIPC-derived SOs. Gray dots represent genes that are not significantly differentially expressed. **e** A Venn diagram shows direct comparisons of the global gene expression profiles between dediHep-derived SOs and FIPC-derived SOs. **f** Representative bright-field and fluorescence images of a recipient mouse colon 3 months after transplantation. Arrowheads indicate GFP⁺ donor-derived engraftments. **g** Co-immunofluorescence staining (GFP with Villin, Muc2, Cdx2, Klf5, EphB2, or Sox9) of recipient mouse colons 3 months after transplantation. **h** Transition of the body weights of mice injected with phosphate-buffered saline (PBS) (control) or dediHep-derived SOs for 7 days after second injection. All data represent the difference from the weights of the mice immediately before the second injection. DNA was stained with DAPI. Scale bars, 200 μm (**a**), 50 μm (**a**; HE staining), 20 μm (**c**, **g**), and 1 cm (**f**). Data represent the mean ± SEM (*n* = 3 independent experiments). Statistical difference was determined by one-way analysis of variance followed by Dunnett's multiple comparison test (**b**), quasi-likelihood F-test (**d**), or two-sided Student's *t* test (**h**). Source data are provided as a Source Data file.

self-renewing cell divisions of ISCs, differentiation into multiple functional cells of the villi, and the release of apoptotic cells into the central lumen[32]. Although FIPCs do not form BOs, they are capable of developing into ISCs that form BOs by serial passaging in the presence of WCENR and subsequent exclusion of Wnt3a and CHIR99021 from the culture medium[29]. To examine whether dediHep-derived SOs could develop into BOs, dediHeps were subjected to 3D culture in the presence of WCENR, followed by removal of Wnt3a and CHIR99021 from the culture medium to promote their spontaneous maturation to BOs. However, unlike FIPC-derived SOs, dediHep-derived SOs did not form BOs during passaging, even after removal of Wnt3a and CHIR99021 from the culture medium (Fig. 6a). Our previous research has demonstrated that, in addition to endogenous *Cdx2* expression, forced *Cdx2* expression was required for the reprogramming of fibroblasts to FIPCs[28]. Thus, endogenous *Cdx2* expression in dediHeps might be insufficient. To examine this possibility, dediHeps were infected with a retrovirus expressing *Cdx2* and subjected to 3D culture (Fig. 6a–c). Forced expression of *Cdx2* promotes the generation of SOs from dediHeps (Supplementary Fig. 8a). After serial passaging in the absence of Wnt3a and CHIR99021, these SOs developed into BOs that were morphologically indistinguishable from those derived from ISCs (Fig. 6a). Changing the additives in the medium from WCENR to ENR is necessary for the efficient formation of BOs even from dediHeps overexpressing *Cdx2* (Supplementary Fig. 8b, c). dediHep-derived BOs could be maintained by repeated passaging in long-term 3D culture (Supplementary Fig. 8d).

Global gene expression analyses revealed that genes specifically expressed in dediHep-derived BOs or ISC-derived BOs, compared with hepatocytes, are similar (Fig. 6d), with more than 97% of the detected genes being expressed similarly in both types of BOs (Fig. 6e). Immunofluorescence analyses revealed that the cells comprising dediHep-derived BOs consistently expressed basolateral E-cad and apical Villin, indicating that they were typical intestinal epithelial cells, mainly enterocytes, that maintain distinctive apicobasal cell polarity (Fig. 6f). These BOs also contained Paneth cells, enteroendocrine cells, and goblet cells, which were respectively characterized by the expression of Lysozyme (Lyz; also known as Lyz1), Chromogranin A (ChgA), and Muc2 (Fig. 6f). Moreover, the expression of ISC-enriched Wnt target proteins, such as Sox9 and EphB2, in BOs were restricted to the CLDs, and cleaved caspase 3 (CC3)-positive apoptotic cells were released into their central lumens (Fig. 6f). These characteristics closely resemble those of ISC-derived BOs (Fig. 6f). Quantitative polymerase chain reaction (qPCR) analyses also revealed that the expression levels of differentiated intestinal epithelial cell markers, such as *Lyz1*, *Chga*, and *Muc2*, and those of dediHep markers, such as *Alb* and *Vim*, significantly increased and decreased, respectively, in BOs derived from exogenous *Cdx2*-expressing dediHeps (Fig. 6g and Supplementary Fig. 9). These data demonstrate that forced *Cdx2* expression promotes the generation of ISC-like cells from dediHeps, which form BOs and give rise to multilineage intestinal epithelial cells.

## dediHeps are partially similar to chemically induced liver progenitors (CLiPs), but differs in many aspects

Small molecule compounds, such as the Rho-associated kinase inhibitor Y-27632, the TGF-β type I receptor inhibitor A-83-01, and CHIR99021, allowed hepatocyte dedifferentiation in serm-free monolayer culture, and the resultant cells were named CLiPs[12]. To compare dediHeps with CLiPs, we cultured adult mouse hepatocytes in accordance with the published protocol and induced hepatocyte dedifferentiation into CLiPs. CLiPs required a couple of months to proliferate sufficiently, while dediHeps needed only a couple of weeks. In fact, the proliferative capacity of CLiPs was significantly lower than that of dediHeps (Supplementary Fig. 10a). CLiPs were morphologically identified as epithelial cells and expressed Alb, CK19, E-cad, and Vim, similar to dediHeps (Fig. 7a), suggesting that CLiPs also have a hybrid E/M phenotype. Global gene expression analyses revealed that expression levels of *Cdh1*, *Cldn3*, *Ocln*, *Vim*, *Prom1*, *Hnf4α*, and *Hnf1α* were similar, but those of *S100a7a*, *Lamb1*, *Mki67*, and *Krt19* were lower, in CLiPs relative to their expression in dediHeps (Fig. 7b). Meanwhile, several genes encoding liver enzymes were highly expressed, and the expression level of *Cdx2* was significantly lower, in CLiPs compared to their expression in dediHeps (Fig. 7b). Dedifferentiation of human hepatocytes into CLiPs[14] also activates the expression of the mesenchymal cell markers *VIM* and *LAMB1*, in addition to *PROM1*, *MKI67*, and *KRT19* (Supplementary Fig. 10b). Thus, CLiPs partially resemble dediHeps with a hybrid E/M phenotype but have more characteristics as hepatocytes than dediHeps.

To assess the relationship among various types of cells analyzed in this study, we investigated the global gene expression profiles in these cells and performed principal component analysis (PCA) (Fig. 7c). PCA revealed that the gene expression signatures of hepatocytes changed widely after 1 day in monolayer culture and were stepwise close to those of dediHeps as the culture progresses. The gene expression signatures of CLiPs were most similar to those of hepatocytes cultured for 1 week. Upon cell-aggregate formation, the gene expression signatures of dediHeps approached those of hepatocytes cultured for a few days in monolayer culture. The formation of SOs in Matrigel 3D culture moved the gene expression signatures of dediHeps away from those of hepatic lineage cells and bring them closer to those of FIPC-derived SOs. In addition, dediHep-derived BOs and ISC-derived BOs occupy a similar dimensional space, indicating a close resemblance between the gene expression signatures of two different types of BOs.

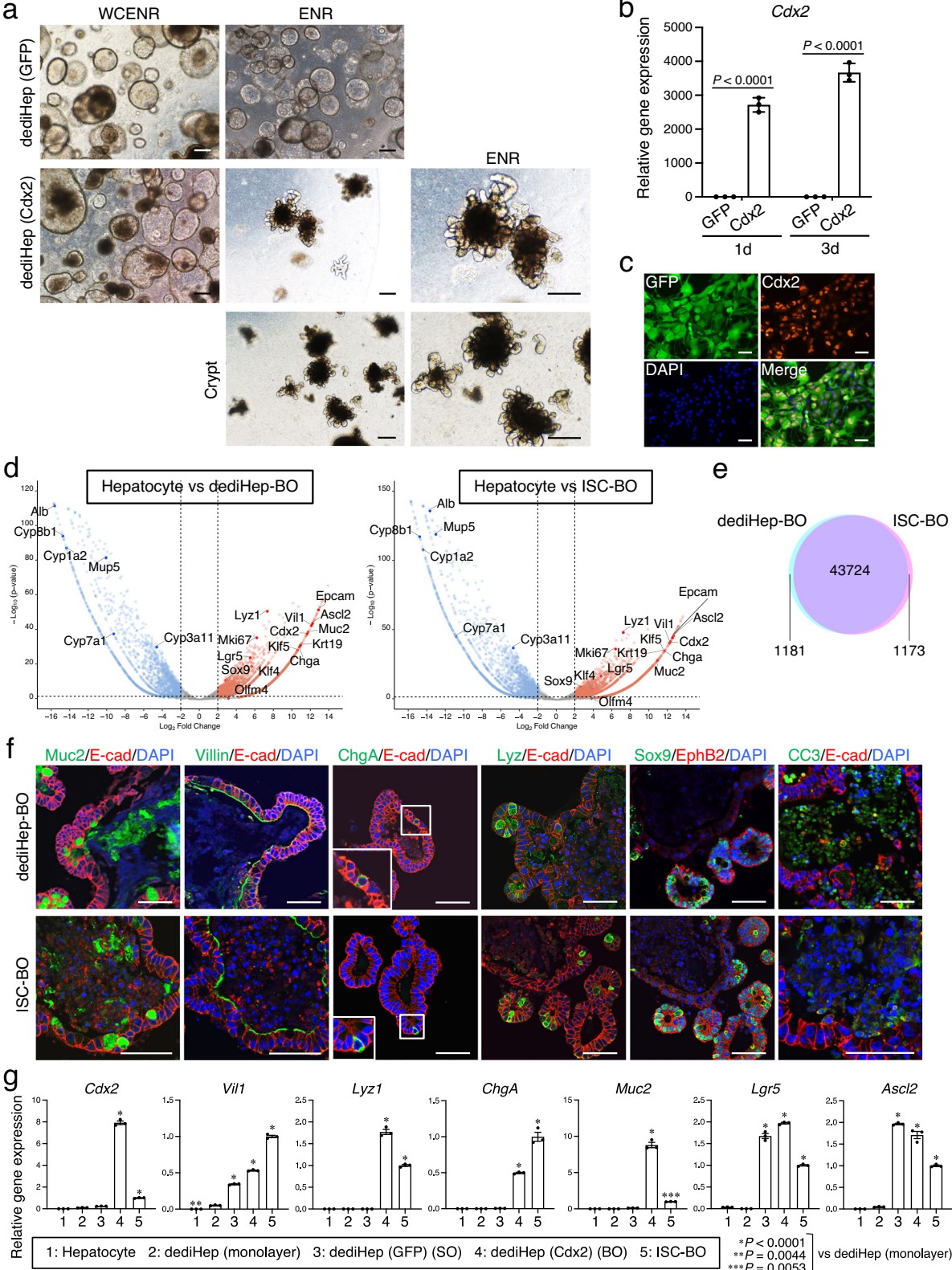

**Hippo inactivation in hepatocytes is fundamental for their conversion to dediHeps**

We next sought to identify the mechanism through which hepatocytes convert to dediHeps. Genes associated with the Hippo signaling pathway were up-regulated in dediHeps and down-regulated in dediHep-derived, re-differentiated hepatocytes (Figs. 2d and 3d). Also, hepatocyte dedifferentiation is induced in vitro and in vivo by activation of Yap, a central

effector molecule of the Hippo signaling pathway[8,10]. Thus, it is suggested that Yap activation is important for the conversion of hepatocytes to dediHeps. Immunofluorescence analyses revealed that Yap protein accumulated in the nucleus of hepatocytes immediately after their dissociation from the liver, which was maintained throughout the period of hepatocyte culture and in the resultant dediHeps (Fig. 8a, b). The expression of the Yap target genes *cysteine rich-61* (*Cyr61*) and *connective*

**Fig. 6 | dediHeps give rise to ISC-like cells following forced expression of *Cdx2*. a** Representative micrographs of 3D cultures of dediHeps transduced with *GFP* or *Cdx2* (with *GFP*). Initially, dediHeps transduced with *GFP* or *Cdx2* were embedded in Matrigel and cultured with WCENR. After passage 5, dediHep-derived SOs and BOs were cultured with ENR to promote the spontaneous transition of SOs to BOs. Crypts were consistently cultured with ENR. **b** qPCR analyses of *Cdx2* expression by dediHeps transduced with *GFP* or *Cdx2* (with *GFP*) 1 day (d) and 3 d after viral infection. All data were normalized to the value for dediHeps transduced with *GFP* at 1 d and are depicted as fold-changes. **c** Co-immunofluorescence staining (GFP with Cdx2) of dediHeps transduced with *Cdx2* (with *GFP*) 1 day after viral infection. **d** Volcano plots depicting differences in gene expression between hepatocytes and dediHep-derived BOs or ISC-derived BOs. Each dot represents one gene. Blue and red dots represent down- and up-regulated genes, respectively, in dediHep-derived BOs or ISC-derived BOs. Gray dots represent genes that are not significantly differentially expressed. **e** A Venn diagram shows direct comparisons of the global gene expression profiles between dediHep-derived BOs and ISC-derived BOs. **f** Co-immunofluorescence staining (E-cad with Muc2, Villin, ChgA, Lyz, or CC3 and Sox9 with EphB2) of dediHep-derived BOs and ISC-derived BOs. Insets, representative ChgA⁺ E-cad⁺ cells. **g** qPCR analyses of expression of *Cdx2* and *Vil1* and marker genes for differentiated intestinal epithelial cells (*Lyz1*, *Chga*, and *Muc2*) and ISCs (*Lgr5* and *Ascl2*) in hepatocytes freshly isolated from adult mouse livers, dediHeps maintained in monolayer culture, SOs formed by dediHeps transduced with *GFP*, BOs formed by dediHeps transduced with *Cdx2* (with *GFP*), and ISC-derived BOs. All data were normalized to the values for ISC-derived BOs and are depicted as fold-changes. DNA was stained with DAPI. Scale bars, 200 μm (**a**) and 50 μm (**c**, **f**). Data represent means ± SEM (*n* = 3 independent experiments). Statistical difference was determined by two-sided Student's *t* test (**b**), quasi-likelihood F-test (**d**), or one-way analysis of variance followed by Dunnett's multiple comparison test (**g**). Source data are provided as a Source Data file.

*tissue growth factor* (*Ctgf*) was significantly increased during culture and maintained in dediHeps (Fig. 8c). Gene set enrichment analysis (GSEA) revealed that Yap-associated genes were up-regulated in dediHeps compared with hepatocytes (Fig. 8d). Meanwhile, Yap immediately disappeared from the nucleus of dediHeps as a result of cell-aggregate formation that induced re-differentiation of dediHeps into growth-arrested functional hepatocytes under 3D culture conditions (Fig. 8e and Supplementary Fig. 11). To synchronize with this, expression levels of *Cyr61* and *Ctgf* also decreased (Fig. 8f).

To examine the importance of Yap activation in hepatocytes, we treated them with verteporfin, an inhibitor of Yap transcriptional activity. This inactivation blocked the conversion of hepatocytes to dediHeps in a concentration-dependent manner in verteporfin (Fig. 8g, h) and led to higher expression of several genes encoding liver enzymes and lower expression of *Vim* and *Prom1* than hepatocytes cultured without verteporfin (Fig. 8i and Supplementary Fig. 12). Since Yap is activated in FIPC-derived SOs and its inhibition impairs SO formation[33], we examined the role of Yap activation in dediHep-derived SOs. Our data demonstrated that dediHep-derived SOs were composed of cells marked with nuclear Yap, and their formation was nearly completely blocked in culture with verteporfin (Supplementary Fig. 13). Thus, Yap activation is critical for the induction of the hybrid E/M phenotype in cultured hepatocytes, their conversion to dediHeps, and differentiation of dediHeps into intestinal epithelial cells.

### Hepatocyte-to-dediHep conversion occurs independently of TGF-β signaling, and Vim is required for the maintenance of dediHeps in culture

Our present data demonstrate that primary hepatocytes acquire a hybrid E/M phenotype in the early stages of culture, which was maintained during and after the conversion of hepatocytes into dediHeps. It is well-known that TGF-β signaling plays an important role in EMT[34]. Thus, to determine whether TGF-β signaling was involved in the conversion of hepatocytes into dediHeps, we isolated hepatocytes from the livers of wild-type mice and *Alb-Cre;Tgfbr2^fl/fl* mice (deficient in the liver-specific TGF-β receptor 2), then cultured them in the presence or absence of TGF-β. We found that TGF-β signaling was toxic to hepatocytes and unnecessary for their conversion to dediHeps (Fig. 9a, b). TGF-β signaling induced EMT by completely suppressing E-cad expression, while Vim expression was induced in cultured hepatocytes both with and without TGF-β activation (Fig. 9c). These results indicate that the mesenchymal phenotype appearing in cultured hepatocytes is induced in a TGF-β-independent manner, and the hybrid E/M cell state is required for the generation of dediHeps.

One of the major mesenchymal features of dediHeps is Vim expression. Thus, we next investigated its role. To this end, we used two kinds of short hairpin RNA (shRNA) targeting *Vim*, designated shVim986 and shVim1183, to reduce *Vim* expression (Fig. 9d). The dediHeps in which *Vim* had been knocked down were morphologically different from, and proliferated much slower than, dediHeps expressing a control shRNA (Fig. 9e, f). Moreover, by knocking down *Vim* expression, the expression levels of *Alb* and *Cdx2* were decreased and increased, respectively, in dediHeps (Fig. 9g), and the number of SOs formed from dediHeps was increased (Fig. 9h). Thus, Vim is essential for the maintenance of dediHeps in culture by inducing their proliferation and interfering intestinal differentiation.

## Discussion

A summary of the findings in this study are shown in Fig. 10. In monolayer culture, adult mouse hepatocytes acquire a hybrid E/M phenotype and dedifferentiate into dediHeps through inactivation of Hippo signaling. The resultant dediHeps exhibit Vim-dependent in vitro propagation, express *Cdx2* through activation of BMP signaling, and differentiate into functional hepatocytes and intestinal epithelial cells under 3D culture conditions. These dediHep-derived hepatocytes and intestinal epithelial cells are capable of reconstituting liver and intestinal epithelial tissues in vivo, respectively, after transplantation into injured organs. Moreover, upon forced expression of *Cdx2*, dediHeps give rise to ISC-like cells that form BOs and differentiate into multilineage intestinal epithelial cells in 3D culture. These findings demonstrate that hepatocytes have an unexpected differentiation potential that can be activated by inducing a hybrid E/M cell state in monolayer culture. The expression of mesenchymal cell markers including *Vim* has been observed in mouse and human hepatocytes in vivo during their conversion to progenitors[7] and CLiPs in vitro, suggesting that the acquisition of a mesenchymal phenotype without losing the epithelial phenotype is a key event in hepatocyte dedifferentiation, and that hepatocytes have high plasticity.

Even though dediHeps express *Cdx2*, forced expression of *Cdx2* is required for the formation of BOs. In the adult intestine, Cdx2 regulates genes distinct from those during embryonic stages, such as *intestinal alkaline phosphatase*, *fatty acid-binding protein 1* and *2*, *lactase*, *sucrase isomaltase*, *meprin A subunit beta*, and *microsomal triglyceride transfer protein*[35]. Our transcriptome analyses revealed that the expression levels of these genes in dediHep-derived BOs were higher than those in dediHep-derived SOs and FIPC-derived SOs and similar to those in ISC-derived BOs. Thus, it is suggested that the expression level of intrinsic *Cdx2* in dediHeps is insufficient to induce the upregulation of adult intestine-specific Cdx2 target genes.

It has been reported that dedifferentiation of adult mouse hepatocytes into CLiPs requires TGF-β inhibition[12]. However, our data show that, although TGF-β activation induces the EMT, TGF-β inhibition is not necessary for the induction of hepatocyte dedifferentiation. CLiPs have some properties closer to hepatocytes than dediHeps, suggesting that TGF-β inhibition works to maintain the characteristics of hepatocytes more effectively. Moreover, we found that hepatocytes adopt a hybrid E/M phenotype in monolayer culture, even in the absence of the TGF-β activation that is typically a strong inducer of mesenchymal

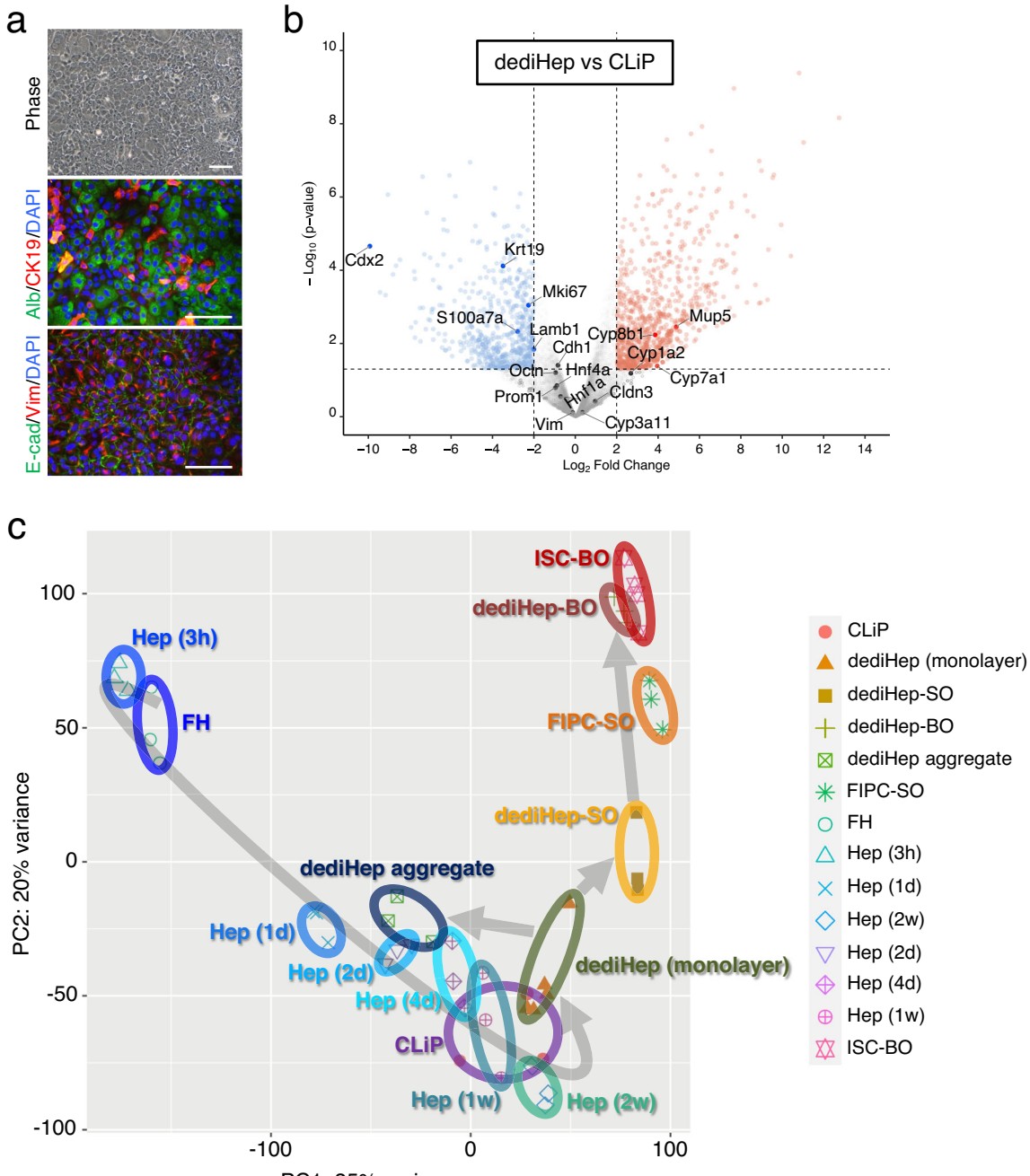

**Fig. 7 | dediHeps are different from CLiPs and give rise to both hepatic and intestinal lineage cells. a** Representative morphologies and fluorescence micrographs of CLiPs expressing Alb, CK19, E-cad, and Vim. DNA was stained with DAPI. Scale bars, 100 μm. **b** Volcano plots depicting differences in gene expression between dediHeps and CLiPs. Each dot represents one gene. Blue and red dots represent genes whose expression are lower and higher in CLiPs than dediHeps, respectively. Gray dots represent genes that are not significantly differentially expressed. Statistical difference was determined by quasi-likelihood F-test. **c** PCA was performed using CEL-seq2 data for freshly isolated hepatocytes (FH), hepatocytes (Hep) in monolayer culture 3 h, 1 day (d), 2 d, 4 d, 1 week (w), and 2 w after plating, dediHeps in monolayer culture, dediHep aggregates, dediHep-derived SOs, FIPC-derived SOs, dediHep-derived BOs, ISC-derived BOs, and CLiPs.

characteristics in epithelial cells. Thus, TGF-β signaling may not be involved in hepatocyte dedifferentiation.

In addition to TGF-β activation, Hippo inhibition is another mechanism that induces an EMT[36]. Our data indicate that Hippo inhibition-induced Yap activation is essential for hepatocyte dedifferentiation. Moreover, Yap inactivation in hepatocyte cultures significantly decreases *Vim* expression, resulting in disruption of the hybrid E/M phenotype. This suggests that the Hippo inhibition characteristic of hepatocytes in monolayer culture functions to induce and maintain the hybrid E/M phenotype in cultured hepatocytes.

In a previous study, we found that expression of *Hnf4α* and *Foxa1*, *Foxa2*, or *Foxa3* converted mouse fibroblasts into hepatocyte-like cells[37]. These induced hepatocyte-like cells (iHepCs) have hepatocyte-specific properties, can be maintained in long-term monolayer cultures, and reconstitute liver tissues after transplantation into injured livers. Meanwhile, iHepCs have been shown to have the potential for differentiation into intestinal epithelial cells that form SOs in 3D culture and reconstitute colonic epithelial tissues after transplantation into a chemically-induced colonic injury model, suggesting that iHepCs can be viewed as induced endoderm progenitors (iEPs)[38]. While *Cdx2* is

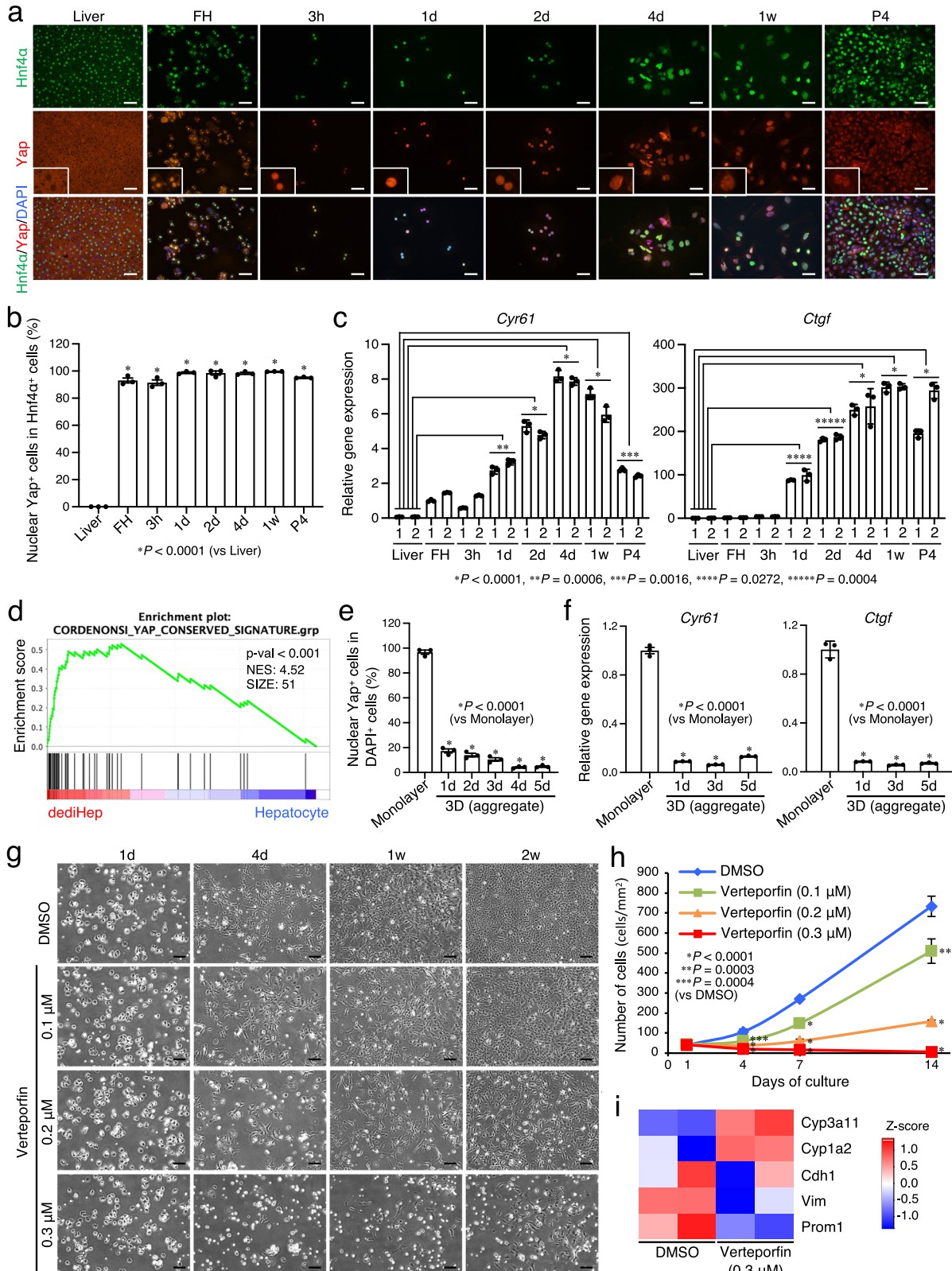

expressed in iHepCs, forced expression of *Gata6* and *Cdx2* in addition to *Hnf4α* and *Foxa3* is required for direct reprogramming of fibroblasts to FIPCs[28]. These induced FIPCs (iFIPCs) can develop into multipotent induced ISC (iISCs) and form BOs under 3D culture conditions, although iHepCs cannot give rise to iISCs and form BOs[28]. Since iHepCs are completely different from iFIPCs, it is interesting that iHepCs can give rise to intestinal epithelial cells. Our present data demonstrate that

dediHeps can differentiate into intestinal epithelial cells, including ISC-like cells, in response to *Cdx2* overexpression. Despite their differences, dediHeps and iHepCs share some common characteristics. Both cell types are capable of propagation in monolayer culture, their hepatic functions are insufficient compared with those of primary hepatocytes, and they become functionally similar to hepatocytes by forming aggregates in 3D culture[23]. These findings suggest that iHepCs

**Fig. 8 | Conversion of hepatocytes to dediHeps requires Yap activation. a** Co-immunofluorescence staining (Hnf4α with Yap) of adult mouse liver tissues, hepatocytes freshly isolated from adult mouse livers (FH), and hepatocytes in monolayer culture 3 h, 1 day (d), 2 d, 4 d, and 1 week (w) after plating and at passage 4 (P4). Insets, subcellular localization of Yap protein. **b** Percentages of Hnf4α⁺ cells positive for nuclear Yap. **c** qPCR analyses of *Cyr61* and *Ctgf* expression in liver tissues and the indicated cells. All data were normalized to the values for freshly isolated hepatocytes (FH)−1 and are depicted as fold-changes. **d** GSEA of whole-transcriptome data from dediHeps and freshly isolated hepatocytes using a set of YAP-associated genes. **e** Immunofluorescence staining of Yap was conducted for dediHeps in monolayer culture and dediHep aggregates at the indicated days (d) after initiation of 3D culture. The graph shows the percentages of cells marked with nuclear Yap in monolayer and 3D cultures of dediHeps. **f** qPCR analyses of *Cyr61* and *Ctgf* expression in dediHeps maintained in monolayer culture and dediHep aggregates 1 d, 3 d, and 5 d after initiation of 3D culture. All data were normalized to the values for dediHeps in monolayer culture and are depicted as fold-changes. **g** Representative micrographs of hepatocyte cultures treated with DMSO or verteporfin (0.1, 0.2, and 0.3 μM) 1 d, 4 d, 1 w, and 2 w after plating. **h** Growth of hepatocyte-derived cells cultured with DMSO or verteporfin (0.1, 0.2, and 0.3 μM). **i** Heatmap image from CEL-seq2 data showing that the expression levels of the indicated genes differ between DMSO- and verteporfin (0.3 μM)-treated dediHeps. DNA was stained with DAPI. Scale bars, 50 μm (**a**) and 200 μm (**g**). Data represent means ± SEM (**b**, **e**, **f**, and **h**) and SD (**c**) (*n* = 3 independent experiments). Statistical difference was determined by one-way analysis of variance followed by Dunnett's multiple comparison test (**b**, **c**, **e**, **f**, and **h**) or permutation test (**d**). Source data are provided as a Source Data file.

maintained in monolayer culture are in a dedifferentiated state and, similar to dediHeps, exhibit high plasticity with the ability to differentiate into intestinal epithelial cells. Thus, this unexpected ability of iHepCs to differentiate into intestinal epithelial cells may be attributed to their plasticity rather than their iEP-related characteristics.

In the present study, we have shown that dediHeps can differentiate into hepatocytes and intestinal epithelial cells. In addition to this unique characteristic, dediHeps might also be able to give rise to cells in other endoderm-derived organs, including lung, stomach, and pancreas, when cultured in suitable conditions. Severe hepatic damage induces intestinal and pancreatic metaplasia in the liver[39,40], suggesting that hepatocyte conversion to endodermal linages can occur under certain circumstances. Further research on dediHeps will advance our understanding of the molecular mechanisms underlying cellular identity and plasticity, which will be important to the development of therapeutic strategies for diseases of the liver and other endoderm-derived organs.

## Methods
### Mice
C57BL/6 (Clea Japan, Tokyo, Japan), *Alb-Cre* (The Jackson Laboratory, Bar Harbor, ME)[41], *Alb-CreER^{T2}* (a gift from Drs. Pierre Chambon and Daniel Metzger)[20], *R26R^{YFP/YFP}* (a gift from Dr. Frank Costantini)[21], *Fah^{-/-}* (RBRC05362) (RIKEN, Japan)[19], NSG (NOD.Cg-*Prkdc^{scid}Il2rg^{tm1Wjl}*/SzJ) (Charles River Laboratories, Wilmington, MA), and *Tgfbr2^{fl/fl}* (a gift from Dr. Jürgen Roes)[42] mice were used in this study. For induction of Cre activity, male mice (8 week-old) were given a single intraperitoneal injection of TM (7.5 mg/mouse; Sigma-Aldrich, St. Louis, MO) dissolved in olive oil (Nacalai Tesque, Kyoto, Japan) at a concentration of 50 mg/mL. TM was injected 2 weeks before isolation of hepatocytes. Mice were housed in groups of 2–4 per cage in a 12-h light/dark cycle (08:00–20:00 light; 20:00–08:00 dark), with controlled room temperature (22 ± 4 °C) and relative humidity (60%). The experiments were approved by the Kyushu University Animal Experiment Committee, and the care of the animals was in accordance with institutional guidelines.

### Cell culture
Hepatocytes were isolated from 10 week-old adult male mouse livers by two-step collagenase digestion[43] and carefully separated from non-parenchymal cells and residual blood cells by repeated centrifugation. Hepatocytes were plated on type I collagen-coated 6-well plates (Iwaki, Tokyo, Japan) at a density of $1 \times 10^5$/well and cultured in our standard medium for hepatic progenitor cells[17–19]. For culture of hepatic progenitor cells, both hepatocyte growth factor (HGF) and EGF were added to the standard medium[17–19]. However, in this study, we only added EGF (20 ng/mL) (Sigma-Aldrich), because HGF affects the induction of hepatocyte differentiation from hepatic progenitor cells[44]. Aggregates were formed from $3 \times 10^3$ dediHeps per well of ultra-low attachment 96-well plates, coated with poly 2-hydroxyethyl methacrylate (Sumitomo Bakelite, Tokyo, Japan) and maintained in a medium used for iHepC aggregation[23]. To differentiate dediHeps into intestinal epithelial cells, $3 \times 10^4$ dediHeps

were embedded in Matrigel (BD Biosciences, San Jose, CA) and cultured in a medium used for the culture of intestinal organoids[28]. Fetal intestinal cells and adult intestinal crypts were isolated from 13.5 days post coitum mouse embryos and 10 week-old adult male mice, respectively, embedded in Matrigel, and cultured to form organoids as described previously[28]. Noggin (100 ng/mL; Peprotech, Cranbury, NJ), verteporfin (0.1, 0.2, and 0.3 μM; Sigma-Aldrich), and TGF-β (10 ng/mL; Peprotech) were used after dilution with water, dimethyl sulfoxide (DMSO) (Nacalai Tesque), and citric acid (Nacalai Tesque), respectively. To knock down *Vim* expression, dediHeps were transduced with a lentiviral vector (CS-RfA-CG; a gift from Dr. Hiroyuki Miyoshi) containing each shRNA (Control shRNA against *LacZ*, 5′-CGCTAAATACTGGCAGGCGTT-3′; shVim986, 5′-GAATGG-TACAAGTCCAAGT-3′[45]; and shVim1183, 5′-AGCTGCTAACTACCAG-GACACTATT-3′) and the gene encoding GFP. Lentivirus production and transduction of cells were carried out as described[23]. 293 T cells were a gift from Dr. Hiroyuki Miyoshi and used to produce recombinant lentiviruses. Dedifferentiaton of adult mouse hepatocytes into CLiPs was induced as described previously[12].

### Immunostaining
Staining of hepatocytes and dediHeps in monolayer cultures was conducted as previously described[37]. For freshly isolated hepatocytes, cells in suspension were adhered to glass slides with Cytospin 4 (Thermo Fisher Scientific, Waltham, MA) before starting immunostaining. Paraffin sections of liver and colonic tissues, dediHep aggregates, dediHep-derived SOs and BOs, and ISC-derived BOs were prepared and analyzed by immunofluorescence and immunohistochemical staining as previously described[28,37]. The primary and secondary antibodies used in this study are listed in Supplementary Table 1.

### mRNA fluorescence in situ hybridization (FISH) assay
dediHeps were plated on 4-well culture slides (Corning, Corning, NY) coated by type I collagen (Nitta Gelatin, Osaka, Japan) at a density of $2 \times 10^5$/well and cultured in our standard medium with EGF for 1 day. Then, FISH was conducted using the RNAscope Multiplex Fluorescent Kit v2 (Advanced Cell Diagnostics, Newark, CA), according to the manufacturer's instructions. FISH probes for *Cdx2* mRNA (RNAscope Probe, Mm-Cdx2) and *DapB* mRNA (RNAscope Negative Control Probe, DapB) (Advanced Cell Diagnostics) were used. TSA Plus tetra-methylrhodamine (Akoya Biosciences, Marlborough, MA) was diluted 1: 750 with a TSA buffer. Following FISH, the same culture slides were used for immunofluorescence staining of E-cad.

### Gene expression analyses
Total RNA was isolated from liver tissues, hepatocytes freshly isolated from adult mouse livers, hepatocytes and dediHeps in monolayer cultures, dediHep aggregates, dediHep-derived SOs, dediHep-derived BOs, and ISC-derived BOs using the ISOGEN II kit (Nippon gene, Tokyo, Japan) and the RNeasy Mini kit (Qiagen, Hilden, Germany) according to the manufacturer's instructions, and cDNAs were synthesized from the

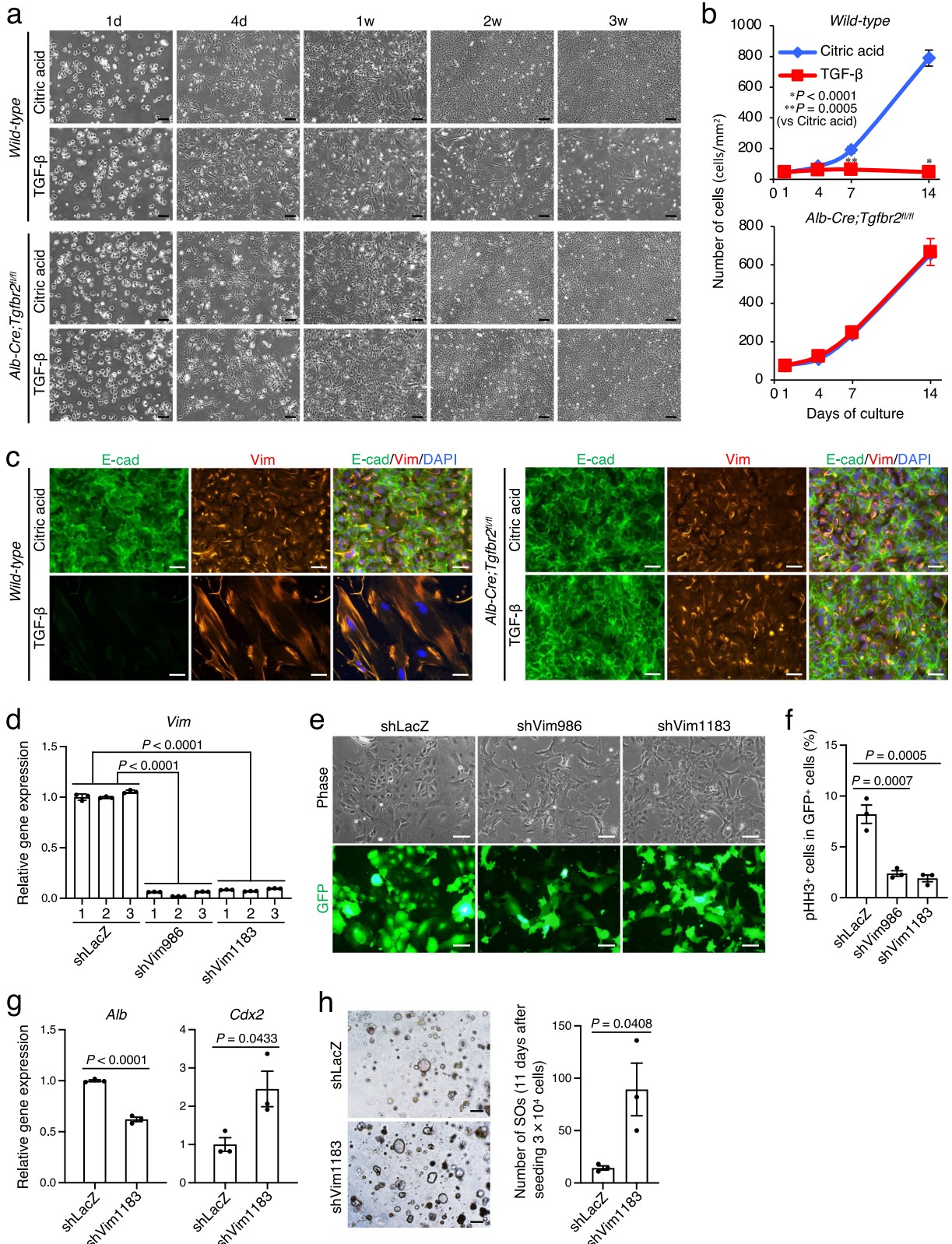

total RNA as previously described[17]. qPCR analyses using TaqMan probes (Applied Biosystems, Waltham, MA) were conducted as previously described[37]. Our assays used TaqMan probes for Cdx2 (Mm01212280_m1), Lyz1 (Mm00657323_m1), Chga (Mm00514341_m1), Muc2 (Mm00458299_m1), Vil1 (Mm00494146_m1), Lgr5 (Mm00438890_m1), and Ascl2 (Mm01268891_g1). TaqMan Gene Expression Assay IDs (Applied Biosystems) are shown in parentheses

following each gene name. As a normalization control, we used TaqMan Rodent GAPDH Control Reagents (Applied Biosystems). To examine the expression of additional genes, qPCR was performed using the THUN-DERBIRD SYBR qPCR Mix (Toyobo, Osaka, Japan) according to the manufacturer's instructions. qPCR primers for *Alb*, *Cyp1a2*, *Cyp3a11*, *Cyp7a1*, *Vim*, *Ki67*, *Ctgf*, *Cyr61*, and *GAPDH* are listed in Supplementary Table 2. The values for *GAPDH* were used as normalization controls.

**Fig. 9 | Roles of TGF-β signaling and Vim in the induction and maintenance of dediHeps in culture. a–c** Hepatocytes isolated from the livers of adult wild-type and *Alb-Cre;Tgfbr2^{fl/fl}* mice cultured with citric acid or TGF-β. **a** Representative micrographs of hepatocyte cultures 1 day (d), 4 d, 1 week (w), 2 w, and 3 w after plating. **b** Growth of hepatocyte-derived cells. **c** Co-immunofluorescence staining (E-cad with Vim) of hepatocyte-derived cells at 2 w in culture. DNA was stained with DAPI. **d** qPCR analysis of *Vim* expression in dediHeps expressing *GFP* with a control shRNA against lacZ (shLacZ), shVim986, or shVim1183. All data were normalized to the values for dediHeps expressing shLacZ (shLacZ-1) and are depicted as fold-changes. **e** Representative morphologies and fluorescence micrographs of GFP⁺ dediHeps expressing shLacZ, shVim986, or shVim1183. **f** Percentages of GFP⁺/sh⁺

(shLacZ, shVim986, or shVim1183) dediHeps that express phospho-histone H3 (pHH3). **g** qPCR analyses of *Alb* and *Cdx2* expression in dediHeps expressing shLacZ or shVim1183. All data were normalized to the values for dediHeps expressing shLacZ and are depicted as fold-changes. **h** Representative morphologies and the number (right graph) of SOs formed by dediHeps expressing shLacZ or shVim1183 at 11 d after initiation of 3D culture. Scale bars, 200 μm (**a**, **h**), 50 μm (**c**), and 100 μm (**e**). Data represent means ± SEM (**b**, **f**, **g**, and **h**) or SD (**d**) (*n* = 3 independent experiments). Statistical difference was determined by two-sided Student's *t* test (**b**, **g**, and **h**) or one-way analysis of variance followed by Dunnett's multiple comparison test (**d**, **f**). Source data are provided as a Source Data file.

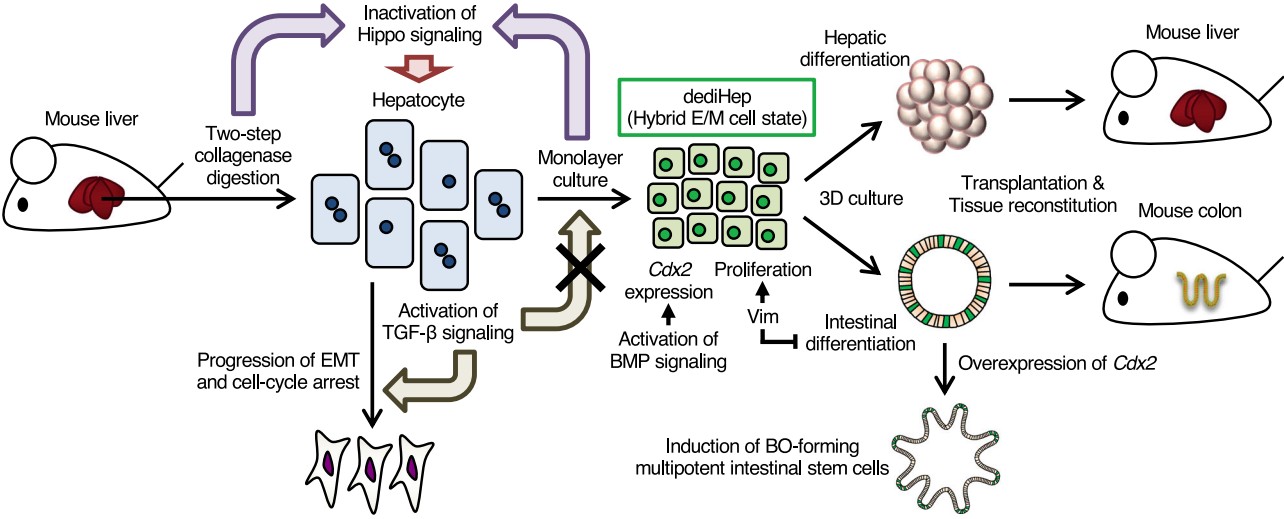

**Fig. 10 | Schema for hepatocyte dedifferentiation and the specific properties of dediHeps.** In monolayer culture, hepatocytes isolated from adult mouse livers dedifferentiate into dediHeps as a result of inactivation of the Hippo signaling pathway. These dediHeps exhibit a hybrid E/M phenotype and Vim-dependent propagation, express *Cdx2* through activation of the BMP signaling pathway, and

give rise to hepatocytes and intestinal epithelial cells under 3D culture conditions, which can respectively reconstitute hepatic tissues and colonic epithelial tissues after transplantation. Forced *Cdx2* expression in dediHeps facilitates their conversion to ISC-like cells that form BOs and give rise to multilineage intestinal epithelial cells.

## Transcriptome analysis

Total RNA was obtained from hepatocytes freshly isolated from adult mouse livers, hepatocytes and dediHeps in monolayer cultures, dediHep aggregates, dediHep-derived SOs, FIPC-derived SOs, dediHep-derived BOs, ISC-derived BOs, CLiPs, and hepatocytes cultured with DMSO or verteporfin (0.3 μM) for 1 week after plating as described above and analyzed by 3′ untranslated region sequencing (CEL-seq2)[46–48]. Calculation of log fold-change (logFC) and identification of differentially expressed genes (FDR < 0.1, 2 < logFC) were performed by edgeR program (version 4.0.2)[49–51] with unique molecular identifier (UMI) count values. KEGG pathway enrichment analyses were performed using DAVID program (v2023q3)[52]. Whole-transcriptome data were visualized using a volcano plot generated by ggVolcanoR[53]. GSEA were performed using GSEA release 4.1.0 and MSigDB release 2.5[54,55]. Heatmaps were generated using ggplot2 program (version 3.4.4)[56] based on normalized UMI count values obtained from iDEP program (version 0.96)[57] with default parameters. All data sets were deposited in the Gene Expression Omnibus (GEO) database under Accession Number GEO: GSE248554. In the analysis of human CLiPs, the logFC between sample pairs were calculated with GEO2R program in the GEO for publicized microarray data (Accession Number GEO: GSE133797).

## Forced expression of *Cdx2* in dediHeps

Production of a retrovirus expressing *Cdx2* and transduction of cells were conducted as described previously[28]. Plat-E cells[58] were a gift from Dr. Toshio Kitamura and used to produce recombinant retroviruses.

## Functional analysis of intestinal epithelial cells

The function of CFTR in the dediHep-derived SOs was evaluated as described previously[59]. Briefly, dediHep-derived SOs were cultured in our intestinal organoid medium[28] without FSK for 24 h, and then 5 μM FSK (Nacalai Tesque) or FSK plus 100 μM CFTRinh-172 (Sigma-Aldrich) were added to the medium. The diameter of 10 spheroids was measured before addition of FSK or FSK plus CFTRinh-172 to the medium, and the diameter of corresponding spheroids was also measured after culture with or without FSK and/or CFTRinh-172 for 24 h.

## Transplantation

Aggregates formed from dediHeps with 10 to 12 passages were cultured for 3 days after initiation of 3D culture. For liver tissue reconstitution, these aggregates were trypsinized, and $1 \times 10^6$ dissociated cells were intrasplenically injected into the livers of *Fah^{-/-}* female mice (8–12 weeks-old) as previously described[19]. Hepatocytes ($1 \times 10^6$) that were freshly isolated from adult mouse livers were also used as donor cells. The *Fah^{-/-}* mice were maintained on drinking water containing 7.5 mg/L NTBC (Swedish Orphan International, Stockholm, Sweden), but treatment was stopped just after transplantation and resumed for 2 weeks when the body weight of recipient mice decreased by 80%. In the analysis of intestinal tissue reconstitution, we employed a DSS-induced colonic injury model using NSG female mice (8–12 weeks-old) and instilled 500–1000 dediHep-derived SOs that were collected from Matrigel using Cell Recovery Solution (Corning) into the colonic lumen at 2 and 5 days after DSS treatment as previously described[28]. dediHeps were marked by infection with a virus expressing *GFP* before

transplantation. The body weight of the mice was monitored daily to evaluate the recovery from colitis for 1 week after second injection of the dediHep-derived SOs.

## Statistics and reproducibility

Statistical significance was analyzed using two-sided Student's *t* test, Dunnett's multiple comparison test, quasi-likelihood F-test, Fisher's exact test, and permutation test. A difference at $P < 0.05$ was considered statistically significant. The significance of an observed enrichment score in GSEA was assessed by comparing it with the set of null enrichment scores computed with randomly assigned phenotypes. Cell and tissue images presented in the figures and Supplementary Figs. were obtained from at least three independent experiments, and representative images are shown.

## Reporting summary

Further information on research design is available in the Nature Portfolio Reporting Summary linked to this article.

## Data availability

All data sets were deposited in the GEO database under Accession Number GEO: GSE248554. The publicly available data sets (Accession Number GEO: GSE133797) were also used in this study. All other data supporting the results presented herein are available within the article and Supplementary Information and from the corresponding author upon reasonable request. A reporting summary for this article is available as a Supplementary Information file. Source data are provided with this paper.

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

## Acknowledgements

We thank Drs. Pierre Chambon, Daniel Metzger, Frank Costantini, Jürgen Roes, Toshio Kitamura, Masafumi Onodera, and Hiroyuki Miyoshi for providing mice and sharing reagents, and Yuuki Honda, Mariko Tasai, Yoshimi Iwasaki, Mitsuhiro Kurata, Kanako Ichikawa, Ryo Ugawa, and Emiko Koba for excellent technical assistance. This work was supported in part by the JSPS KAKENHI (Grant Numbers: JP18H06069 and JP21K18039 to S.M.; JP23K11851 to K.H.; JP17H05623, JP18H05102, JP19H01177, JP19H05267, JP20H05040, JP21K19916, JP22H05634, JP22H04698, and JP22H00592 to A.S.), the Program for Basic and Clinical Research on Hepatitis of the Japan Agency for Medical Research and Development (AMED) (JP23fk0210116 to A.S.), the Research Center Network for Realization of Regenerative Medicine of AMED (JP23bm1123005 to A.S.), the MEXT Promotion of Development of a Joint Usage/Research System Project: Coalition of Universities for Research Excellence Program (JPMXP1323015486 to Y.O. and A.S.), the Medical Research Center Initiative for High Depth Omics (to S.M., K.H., Y.O., and A.S.), the Takeda Science Foundation (to S.M. and A.S.), the Uehara Memorial Foundation (to S.M., K.H., and A.S.), the Kato Memorial Trust for Nambyo Research (to A.S.), the Suzuken Memorial Foundation (to A.S.), the Naito Foundation (to S.M. and A.S.), and the Shinnihon Foundation of Advanced Medical Treatment Research (to S.M.).

## Author contributions

S.M., S.T., and T.I. performed experiments, collected data, and conducted data analyses. K.I., S.K., J.Y., and S.S. supported in vitro and in vivo experiments and data analyses. S.M., S.T., K.H., and Y.O. performed CEL-seq2 and analyzed the data obtained from CEL-seq2. A.S. contributed to the conception, design, and overall project management and wrote the paper.

## Competing interests

The authors declare no competing interests.
