## [Peer Review File · Nature Communications]

Hepatocytes differentiate into intestinal epithelial cells through a hybrid epithelial/mesenchymal cell state in cultureREVIEWER COMMENTS

Reviewer #1 (Remarks to the Author):

Miura and colleagues use an in vitro system to show that primary mouse hepatocytes undergo a de-differentiation via a hybrid epithelial/mesenchymal phenotype, which can be re-differentiated when cultured in 3D and rescue mouse livers in the FAH mouse model. Furthermore, they report that a subset of these de-differentiated hepatocytes (dediHeps) can form intestinal epithelial cell organoids in vitro. While the data is intriguing, I feel that overall it is lacking in detail. My comments are below.

1. The mesenchymal phenotype needs to be more carefully characterized. I would like to see more than just protein staining. Better to have transcriptomic data over time to characterize how YFP+ hepatocytes are changing in 2D culture.
2. I'd like to see more characterization of the re-differentiation of dediHeps in 3D culture. What's the time course of this phenotypic change in 3D culture? Are there intermediate cell states? Given the heterogeneity within dediHeps shown later in the paper, do all dediHeps re-differentiate or is there only a subpopulation that have this capacity? How long do dediHeps maintain re-differentiation potential?
3. More details are needed for the FAH transplantation studies. No controls (using freshly isolated hepatocytes) were included to allow comparison of repopulation efficacy. Controls would also be important to determine if the repopulated FAH+ cells maintain any of the mesenchymal phenotype observed in dediHeps.
4. For dediHep differentiation into intestinal epithelial cells, the analysis done is not detailed enough to show that the spheroidal organoids are intestinal epithelial cells. I would like to see transcriptome analysis and compare with the FIPC-derived SOs.
5. The methods indicate dediHeps were cultured with adult intestinal crypts. Can the authors ensure that the cells that grow as intestinal cells are from dediHeps and not the co-cultured cells? Do these cells all express YFP? Can you also rule out the possibility of cell fusion? Do the engrafted intestinal cells express YFP?
6. What is the time course of Yap disappearance from dediHeps? Is Yap expression in all hepatocytes or is there heterogeneity? Does Yap signaling get downregulated in cell-aggregates? Does down-regulation of Yap maintain hepatocyte phenotype in vitro? Does this prevent intestinal fate in vitro?

Minor comment:

In the introduction, authors state that the references cited "suggest that hepatocyte dedifferentiation occurs in vivo as well as in vitro in the injured liver and contributes to liver regeneration". However, the references cited are all in vitro papers. The authors later state that "induction of hepatocyte dedifferentiation is considered a promising therapeutic strategy for the treatment of liver diseases", however there is no support provided for this.

Reviewer #2 (Remarks to the Author):

In their manuscript, "Hepatocytes differentiate into intestinal epithelial cells through a hybrid epithelial/mesenchymal cell state in culture", authors Miura et al demonstrate that dedifferentiated hepatocytes have differentiation potential into intestinal epithelial cells. The authors use both in vivo transplantation model and in vitro organoid culture model to generate intestinal epithelial cells from dediHeps. They also show that hepatocyte-to-dediHep conversion is independent of TGF- β signaling, but require Yap activation. The findings are unexpected and represent an important advancement for the study of hepatocytes plasticity. However, several major and minor points were found upon review of the manuscript which could strengthen the study.

Major points:

1. What is the efficiency of spherical organoids and/or colonic tissues formation derived from Cdx2 negative, Cdx2 positive and Cdx2 over-expressed dediHeps? Figure 5A gives an impression that forced expression of Cdx2 reduces spherical organoids formation.
2. After forced expression of Cdx2 in dediHeps, is the WCENR treatment require for the formation of budding organoid?
3. To solidate the functional role of Yap on dediHeps conversion, experiments with Yap knockout/knockdown, or a dose dependent effect of Yap inhibition by Verteporfi should be provided.

Minor points:

1. It is unclear how long/how many passages the budding organoid can be maintained in the absence of Wnt3a and CHIR99021.
2. Paneth cells normally locate at the bottom of crypts/buds in organoid culture. Is this pattern preserved in dediHeps-derived intestinal organoids. (Figure 5D)
3. All analyzed genes are changed at the same direction in Figure 5E. Adding hepatocyte/ dediHeps markers can serve as technical controls.
4. "In addition to this unique characteristic, dediHeps may also give rise to cells in other endoderm-derived organs, including lung, stomach, and pancreas, when cultured in suitable conditions" this sentence seems speculative.

Reviewer #3 (Remarks to the Author):

Major revision.

The manuscript from Miura et al reported that hepatocyte organoids possess an unexpected differentiation potential into intestinal epithelial cells. They also reported a hybrid epithelial/mesenchymal cell state during hepatocyte dedifferentiation. While these findings are interesting; however, given the heterogeneity of the culture, it is questionable whether the data provided were adequate to support their conclusions. A better characterization of this culture would greatly support the main conclusions.

Major concerns:

1. The presence of the hybrid E/M state in dediHep should be validated by additional criteria, such as molecular profiles. The process of hybrid E/M formation should be addressed experimentally.
2. Heterogeneity should be further described in dediHep-spherical organoids (dediHep-SOs). The proportion of Cdx2 expressed in dediHep-SOs should be determined. The identity of Cdx2- cells in dediHep-SOs should be rigorously evaluated. The authors used different conditions to culture 2D and 3D cells, a comparison of transcriptional profiling of these cells is required. In addition, these cells should be compared with profiles of published intestine organoids.
3. At least, some integration and therapeutic effect of Cdx2+ cells in vivo should be demonstrated.
4. The conclusion of “Yap activation is critical for hepatocyte conversion to dediHeps” cannot be drawn via cell numbers and Vim expression. The transcriptome analysis of cell states is required. Do Yap and Vim control the conversion between dediHeps and intestinal-like cells? The underlying mechanism is not fully explained.
5. Given dedifferentiation of cultured mouse hepatocytes and human hepatocytes has been published previously, the authors are encouraged to compare their cells with published mouse hepatocytes head-to-head (CLiP cells, Takeshi Katsuda et al., Cell Stem Cell, 2017) and re-analyzed the published human data to find whether their conclusions stand in human cells.

Minor concerns:

1. The hepatocyte culture system, the maximum number of passages and passages used for liver transplantation should be described.
2. Figure 4B: Detailed characterization of dediHep-SOs is required, such as liver hepatic/biliary markers, and the functional characterization of intestinal epithelial cells.
3. Figure 4C: the proportion of intestinal epithelial markers in dedihep-SOs? The negative control for the immunostaining is required.
4. Figure 4E: Controls of cdx2-negative cells from the same culture should be included.
5. Figure 5D, 5E: Positive control of intestinal epithelium markers should be included.

Response to the Reviewers

Reviewer #1:

Miura and colleagues use an in vitro system to show that primary mouse hepatocytes undergo a de-differentiation via a hybrid epithelial/mesenchymal phenotype, which can be re-differentiated when cultured in 3D and rescue mouse livers in the FAH mouse model. Furthermore, they report that a subset of these de-differentiated hepatocytes (dediHeps) can form intestinal epithelial cell organoids in vitro. While the data is intriguing, I feel that overall it is lacking in detail. My comments are below.

Response: Thank you very much for your kind comments regarding our manuscript. Your comments were highly insightful and enabled us to greatly improve the quality of our manuscript. We have revised the manuscript to address all of the concerns that you raised, and our point-by-point responses to your comments are shown below.

Major comment:

1. The mesenchymal phenotype needs to be more carefully characterized. I would like to see more than just protein staining. Better to have transcriptomic data over time to characterize how YFP+ hepatocytes are changing in 2D culture.

Response: In accordance with your comment, we analyzed temporal gene expression changes during hepatocyte-to-dediHep conversion in monolayer culture. The obtained transcriptome data showed that the expression of genes encoding liver enzymes began to decrease after 1 day in monolayer culture, followed by subsequent increases in the expression of *MKi67*, *Krt19*, *Prom1*, and mesenchymal cell markers (Fig. 2e).

2. I'd like to see more characterization of the re-differentiation of dediHeps in 3D culture. What's the time course of this phenotypic change in 3D culture? Are there intermediate cell states? Given the heterogeneity within dediHeps shown later in the paper, do all dediHeps re-differentiate or is there only a subpopulation that have this capacity? How long do dediHeps maintain re-differentiation potential?

Response: In accordance with your comment, we observed the process of dediHeps forming aggregates in 3D culture and illustrated representative micrographs in Fig. 3a. These micrographs showed that almost all of dediHeps seeded in the well contributed to cell-aggregate formation. We have shown that only 10-15% of dediHeps express Cdx2 in the previous version of this study. However, we noticed during the revision experiments that the concentration of antibodies for detecting Cdx2 was low. A five-fold increase in the concentration of antibody increased the number of cells detected as Cdx2-positive cells, in which we found dediHeps that were strongly or weakly positive for Cdx2. The percentage of dediHeps strongly expressing Cdx2 is close to the percentage of Cdx2-positive cells shown in the previous version of this study, suggesting that we could previously detect only cells that strongly expressed Cdx2. In the revision experiments, we also conducted fluorescence in situ hybridization assay, which revealed that an even greater number of dediHeps expressed mRNA for Cdx2. Our data showed that more than 60% and 90% of dediHeps expressed Cdx2 and its mRNA, respectively, in dediHeps (Fig. 4a and b). In addition, Cdx2 was co-expressed with Alb in dediHeps (Fig. 4c) and gradually disappeared from dediHeps during their re-differentiation into hepatocytes (Supplementary Fig. 5). Taken together, our data demonstrate that dediHeps are a nearly homogeneous cell population with respect to the expression of *Cdx2*, and almost all of cells contribute to cell-aggregate formation and re-differentiate into hepatocytes in 3D culture.

qPCR analyses revealed that expression of genes encoding liver enzymes, such as *Cyp1a2*, *Cyp3a11*, and *Cyp7a1*, was increased, but expression of *Vim* and *Ki67* was decreased, in aggregates, and these changes in gene expression were promptly induced starting from 1 day of 3D culture (Supplementary Fig. 2). Thus, it is suggested that dediHeps are unlikely to re-differentiate into hepatocytes through an intermediate state. Moreover, we also examined how long dediHeps can maintain re-differentiation potential. To this end, dediHeps that have undergone 20 passages in long-term monolayer culture were used for cell-aggregate formation in 3D culture. Our data showed that the hepatic maturation of dediHep aggregates could be induced from these dediHeps even after a prolonged period of monolayer culture (Supplementary Fig. 3).

3. More details are needed for the FAH transplantation studies. No controls (using freshly isolated hepatocytes) were included to allow comparison of repopulation efficacy. Controls would also be important to determine if the repopulated FAH⁺ cells maintain any of the mesenchymal phenotype observed in dediHeps.

Response: In accordance with your comment, we transplanted hepatocytes freshly isolated from adult mouse livers into the liver of *Fah*^{-/-} mice and compared tissue reconstitution capacity of these hepatocytes and that of cells dissociated from dediHep aggregates. Our data showed that tissue reconstitution capacity in dediHep-derived hepatocytes was similar to that of freshly isolated hepatocytes (Fig. 3f). Moreover, dediHep-derived *Fah*-positive hepatocytes did not express the mesenchymal cell markers Vim and α SMA, similar to hepatocytes engrafted in the liver of *Fah*^{-/-} mice (Supplementary Fig. 4).

4. For dediHep differentiation into intestinal epithelial cells, the analysis done is not detailed enough to show that the spheroidal organoids are intestinal epithelial cells. I would like to see transcriptome analysis and compare with the FIPC-derived SOs.

Response: In accordance with your comment, we performed global gene expression analyses for dediHep-derived SOs and FIPC-derived SOs. FIPCs were isolated from 13.5 days post coitum mouse embryos and cultured to form SOs. Our data showed that genes specifically expressed in dediHep-derived SOs or FIPC-derived SOs, compared with hepatocytes, are similar (Fig. 5d), and more than 97% of the detected genes were similarly expressed in both types of SOs (Fig. 5e).

5. The methods indicate dediHeps were cultured with adult intestinal crypts. Can the authors ensure that the cells that grow as intestinal cells are from dediHeps and not the co-cultured cells? Do these cells all express YFP? Can you also rule out the possibility of cell fusion? Do the engrafted intestinal cells express YFP?

Response: We sincerely apologize for any ambiguity in our explanation and any misunderstanding about our experimental methods. In this study, we have never co-cultured dediHeps with intestinal epithelial cells. To avoid misunderstanding, we have rewritten the description of the method. The new description is as follows: “To differentiate dediHeps into intestinal epithelial cells, 3×10^4 dediHeps were embedded in Matrigel and cultured in a medium used for the culture of intestinal organoids. Fetal intestinal cells and adult intestinal crypts were isolated from 13.5 days post coitum mouse embryos and 10 week-old adult mice, respectively, embedded in Matrigel, and cultured

to form organoids as described previously.”

6. What is the time course of Yap disappearance from dediHeps? Is Yap expression in all hepatocytes or is there heterogeneity? Does Yap signaling get downregulated in cell-aggregates? Does down-regulation of Yap maintain hepatocyte phenotype in vitro? Does this prevent intestinal fate in vitro?

Response: In accordance with your comment, we conducted immunofluorescence staining of Yap for dediHeps in monolayer culture and dediHep aggregates at 1, 2, 3, 4, and 5 days after initiation of 3D culture. Our data showed that Yap immediately disappeared from the nucleus of dediHeps as a result of cell-aggregate formation under 3D culture conditions (Fig. 8e and Supplementary Fig. 11). To synchronize with this, expression levels of the Yap target genes *Cyr61* and *Ctgf* also decreased (Fig. 8f). Thus, it is suggested that YAP signaling got down-regulated in dediHep aggregates, while a small number of Yap-positive cells remained in the aggregates.

As shown in Fig. 8a and b, Yap protein accumulated in the nucleus of almost all of hepatocytes that were freshly isolated from adult mouse livers and cultured in monolayer culture and almost all of dediHeps maintained in monolayer culture. In primary culture of hepatocytes, Yap inactivation by the Yap inhibitor verteporfin led to higher expression of several genes encoding liver enzymes and lower expression of *Vim* and *Prom1* than hepatocytes cultured without verteporfin (Fig. 8i and Supplementary Fig. 12), suggesting that Yap inactivation could partially maintain a phenotype of hepatocytes in monolayer culture. Moreover, dediHep-derived SOs were composed of cells marked with nuclear Yap, and their formation was nearly completely blocked in 3D culture with verteporfin (Supplementary Fig. 13). Thus, Yap activation is required for differentiation of dediHeps into intestinal epithelial cells.

Minor comment:

In the introduction, authors state that the references cited “suggest that hepatocyte dedifferentiation occurs in vivo as well as in vitro in the injured liver and contributes to liver regeneration”. However, the references cited are all in vitro papers. The authors later state that “induction of hepatocyte dedifferentiation is considered a promising therapeutic strategy for the treatment of liver diseases”, however there is no support

provided for this.

Response: In accordance with your comment, we decided to remove the sentence “These data suggest that hepatocyte dedifferentiation occurs *in vivo* as well as *in vitro* in the injured liver and contributes to liver regeneration.” from the text. Also, we have corrected the sentence noted as follows: “Induction of hepatocyte dedifferentiation will be a possible therapeutic strategy for the treatment of liver diseases”.

Reviewer #2:

In their manuscript, “Hepatocytes differentiate into intestinal epithelial cells through a hybrid epithelial/mesenchymal cell state in culture”, authors Miura et al demonstrate that dedifferentiated hepatocytes have differentiation potential into intestinal epithelial cells. The authors use both in vivo transplantation model and in vitro organoid culture model to generate intestinal epithelial cells from dediHeps. They also show that hepatocyte-to-dediHep conversion is independent of TGF- β signaling, but require Yap activation. The findings are unexpected and represent an important advancement for the study of hepatocytes plasticity. However, several major and minor points were found upon review of the manuscript which could strengthen the study.

Response: Thank you very much for your kind comments regarding our manuscript. Your comments were highly insightful and enabled us to greatly improve the quality of our manuscript. We have revised the manuscript to address all of the concerns that you raised, and our point-by-point responses to your comments are shown below.

Major points:

1. What is the efficiency of spherical organoids and/or colonic tissues formation derived from Cdx2 negative, Cdx2 positive and Cdx2 over-expressed dediHeps? Figure 5A gives an impression that forced expression of Cdx2 reduces spherical organoids formation.

Response: We have shown that only 10-15% of dediHeps express Cdx2 in the previous version of this study. However, we noticed during the revision experiments that the

concentration of antibodies for detecting Cdx2 was low. A five-fold increase in the concentration of antibody increased the number of cells detected as Cdx2-positive cells, in which we found dediHeps that were strongly or weakly positive for Cdx2. The percentage of dediHeps strongly expressing Cdx2 is close to the percentage of Cdx2-positive cells shown in the previous version of this study, suggesting that we could previously detect only cells that strongly expressed Cdx2. In the revision experiments, we also conducted fluorescence in situ hybridization assay, which revealed that an even greater number of dediHeps expressed mRNA for Cdx2. Our data showed that more than 60% and 90% of dediHeps expressed Cdx2 and its mRNA, respectively, in dediHeps (Fig. 4a and b), suggesting that dediHeps are a nearly homogeneous cell population with respect to the expression of *Cdx2*. Thus, in the revision experiments, we compared the efficiency of SO formation from dediHeps (Cdx2-positive) and exogenous *Cdx2*-expressing dediHeps. Our data showed that forced expression of *Cdx2* promotes the generation of SOs from dediHeps at 11 days after initiation of 3D culture with WCENR (Supplementary Fig. 8a). In the previous Fig. 5a, we represented the micrograph that shows both SOs and BOs formed from dediHeps overexpressing *Cdx2* after several passages in culture with WCENR, although the number of BOs was very few. Thus, we replaced this micrograph with a representative one (Fig. 6a). It is speculated that forced expression of *Cdx2* might also promote colonic tissue reconstitution upon transplantation. However, as for the efficiency of colonic tissue formation, quantitative comparative experiments were considered difficult due to the large individual differences in the degree of DSS-induced colonic injury and transplantation efficiency.

In the revision, we showed that almost all of dediHeps express *Cdx2* in monolayer culture. Thus, it was suggested that something inducing the expression of *Cdx2* was included in the dediHep monolayer cultures. To address this issue, we analyzed temporal gene expression changes during hepatocyte-to-dediHep conversion in monolayer culture. The obtained transcriptome data showed that expression of *BMP4*, *BMP7*, and the BMP target genes *Id1* and *Id2* was increased in dediHeps (Fig. 4d). In the development, BMP signaling is essential for mid/hindgut specification from definitive endoderm, inducing *Cdx2* expression and the fate of intestine (McCracken et al., *Nature*, 2014; Davenport et al., *Stem Cells*, 2016). Also, ectopic expression of *Id2* in the developing stomach induces *Cdx2*-positive intestinal epithelial cells in the gastric epithelia (Mori et al., *Mol. Cell. Biol.*, 2018). Thus, our data suggested that BMP signaling was activated and critical for *Cdx2* expression, in dediHeps. To test this hypothesis, we cultured dediHeps with the BMP antagonist Noggin. Our data showed that inhibition of BMP signaling by Noggin significantly suppressed the expression level of *Cdx2* in dediHeps (Fig. 4e). These data

demonstrate that dediHeps express *Cdx2* through activation of BMP signaling in monolayer culture.

2. After forced expression of Cdx2 in dediHeps, is the WCENR treatment required for the formation of budding organoid?

Response: In accordance with your comment, we examined whether the WCENR treatment is still required for BO formation from exogenous *Cdx2*-expressing dediHeps. Our data showed that changing the additives in the medium from WCENR to ENR is necessary for the efficient formation of BOs even from dediHeps overexpressing *Cdx2* (Supplementary Fig. 8b and c).

3. To solidate the functional role of Yap on dediHeps conversion, experiments with Yap knockout/knockdown, or a dose dependent effect of Yap inhibition by Verteporfi should be provided.

Response: In accordance with your comment, we treated hepatocytes with DMSO or verteporfin (0.1, 0.2, and 0.3 μ M) for 1 day, 4 days, 1 week, and 2 weeks after plating. Our data showed that Yap inactivation blocked the conversion of hepatocytes to dediHeps in a concentration-dependent manner in verteporfin (Fig. 8g and h). Since the knockdown efficiency of the designed Yap shRNAs was low, verteporfin was used to inhibit the Yap activation.

Minor points:

1. It is unclear how long/how many passages the budding organoid can be maintained in the absence of Wnt3a and CHIR99021.

Response: In accordance with your comment, we continued the passages of dediHep-derived BOs in the absence of Wnt3a and CHIR99021. Our data showed that dediHep-derived BOs could be maintained by repeated passaging in long-term 3D culture, and we could observe typical BOs similar to ISC-derived BOs at least at passage 13 after initiation of 3D culture (Supplementary Fig. 8d).

2. *Paneth cells normally locate at the bottom of crypts/buds in organoid culture. Is this pattern preserved in dediHeps-derived intestinal organoids. (Figure 5D)*

Response: Yes. Lysozyme-positive Paneth cells located at the crypt-like domains of dediHep-derived BOs and ISC-derived BOs (Fig. 6f). To avoid misunderstanding, we replaced the micrograph shown in the previous Fig. 5d to a suitable one (Fig. 6f).

3. *All analyzed genes are changed at the same direction in Figure 5E. Adding hepatocyte/dediHeps markers can serve as technical controls.*

Response: In accordance with your comment, we analyzed the expression levels of the dediHep markers *Alb* and *Vim* in hepatocytes freshly isolated from adult mouse livers, dediHeps maintained in monolayer culture, dediHep-derived SOs and BOs, and ISC-derived BOs by qPCR. Our data showed that the expression levels of *Alb* and *Vim* significantly decreased in dediHep-derived SOs and BOs, compared with dediHeps in monolayer culture (Supplementary Fig. 9).

4. *“In addition to this unique characteristic, dediHeps may also give rise to cells in other endoderm-derived organs, including lung, stomach, and pancreas, when cultured in suitable conditions” this sentence seems speculative.*

Response: We totally agreed with this comment from yours. We have corrected the sentence noted as follows: “In addition to this unique characteristic, dediHeps might also be able to give rise to cells in other endoderm-derived organs, including lung, stomach, and pancreas, when cultured in suitable conditions”.

Reviewer #3:

The manuscript from Miura et al reported that hepatocyte organoids possess an unexpected differentiation potential into intestinal epithelial cells. They also reported a

hybrid epithelial/mesenchymal cell state during hepatocyte dedifferentiation. While these findings are interesting; however, given the heterogeneity of the culture, it is questionable whether the data provided were adequate to support their conclusions. A better characterization of this culture would greatly support the main conclusions.

Response: Thank you very much for your kind comments regarding our manuscript. Your comments were highly insightful and enabled us to greatly improve the quality of our manuscript. We have revised the manuscript to address all of the concerns that you raised, and our point-by-point responses to your comments are shown below.

Major concerns:

1. The presence of the hybrid E/M state in dediHep should be validated by additional criteria, such as molecular profiles. The process of hybrid E/M formation should be addressed experimentally.

Response: In accordance with your comment, we analyzed temporal gene expression changes during hepatocyte-to-dediHep conversion in monolayer culture. The obtained transcriptome data showed that the expression of genes encoding liver enzymes began to decrease after 1 day in monolayer culture, followed by subsequent increases in the expression of *MKi67*, *Krt19*, *Prom1*, and mesenchymal cell markers (Fig. 2e).

2. Heterogeneity should be further described in dediHep-spherical organoids (dediHep-SOs). The proportion of Cdx2 expressed in dediHep-SOs should be determined. The identity of Cdx2- cells in dediHep-SOs should be rigorously evaluated. The authors used different conditions to culture 2D and 3D cells, a comparison of transcriptional profiling of these cells is required. In addition, these cells should be compared with profiles of published intestine organoids.

Response: In accordance with your comment, we investigated the proportion of Cdx2-positive cells in dediHep-derived SOs. As shown in Fig. 5c, about 95% of cells that formed dediHep-derived SOs expressed Cdx2. In addition, about 95% and 98% of cells forming these SOs were also positive for Villin and Sox9, respectively (Fig. 5c). These data demonstrate that dediHep-derived SOs are composed of a nearly homogeneous cell

population, such as intestinal epithelial cells that express Cdx2, Villin, and Sox9. Moreover, we investigated the gene expression signatures of dediHeps in monolayer (2D) culture and dediHep-derived SOs in 3D culture and performed principal component analysis (PCA). PCA revealed that the formation of SOs in Matrigel 3D culture moved the gene expression signatures of dediHeps away from those of hepatic lineage cells and bring them closer to those of FIPC-derived SOs (Fig. 7c). FIPCs were isolated from 13.5 days post coitum mouse embryos and cultured to form SOs. In addition, we performed global gene expression analyses for dediHep-derived SOs and FIPC-derived SOs. Our data showed that genes specifically expressed in dediHep-derived SOs or FIPC-derived SOs, compared with hepatocytes, are similar (Fig. 5d), and more than 97% of the detected genes were similarly expressed in both types of SOs (Fig. 5e).

3. At least, some integration and therapeutic effect of Cdx2+ cells in vivo should be demonstrated.

Response: We have injected dediHep-derived SOs that are composed of Cdx2-positive intestinal epithelial cells into the colons of immunodeficient NSG mice with DSS-induced acute colitis. The donor cells were capable of integrating into and reconstituting colonic epithelial tissues 3 months after injection (Fig. 5f and g). Moreover, we found that the body weights of mice transplanted with dediHep-derived SOs recovered faster than those of control mice (Fig. 5h). These data demonstrate that cells composing dediHep-derived SOs are able to morphologically and functionally repopulate colonic epithelial tissues *in vivo*.

4. The conclusion of “Yap activation is critical for hepatocyte conversion to dediHeps” cannot be drawn via cell numbers and Vim expression. The transcriptome analysis of cell states is required. Do Yap and Vim control the conversion between dediHeps and intestinal-like cells? The underlying mechanism is not fully explained.

Response: In accordance with your comment, we investigated the roles of Yap and Vim in the induction and differentiation of dediHeps in more detail. As a result of global gene expression analyses, we found that genes associated with the Hippo signaling pathway were up-regulated in dediHeps and down-regulated in dediHep-derived, re-differentiated hepatocytes (Fig. 2d and 3d). To examine the importance of Yap activation in hepatocytes,

we treated them with the Yap inhibitor verteporfin. This inhibitor blocked the conversion of hepatocytes to dediHeps in a concentration-dependent manner (Fig. 8g and h). Also, Yap inactivation led to higher expression of several genes encoding liver enzymes and lower expression of *Vim* and *Prom1* than hepatocytes cultured without verteporfin (Fig. 8i and Supplementary Fig. 12), suggesting that Yap inactivation could partially maintain a phenotype of hepatocytes in monolayer culture. Since Yap is activated in FIPC-derived SOs and its inhibition impairs SO formation (Pikkupeura et al., *Sci Adv*, 2023), we examined the role of Yap activation in dediHep-derived SOs. Our data demonstrated that dediHep-derived SOs were composed of cells marked with nuclear Yap, and their formation was nearly completely blocked in culture with verteporfin (Supplementary Fig. 13). Thus, Yap activation is critical for the induction of the hybrid E/M phenotype in cultured hepatocytes, their conversion to dediHeps, and differentiation of dediHeps into intestinal epithelial cells. In addition, we found that, by knocking down *Vim* expression, the expression levels of *Alb* and *Cdx2* were decreased and increased, respectively, in dediHeps (Fig. 9g), and the number of SOs formed from dediHeps was increased (Fig. 9h). Thus, *Vim* is essential for the maintenance of dediHeps in culture by not only inducing their proliferation but also interfering intestinal differentiation.

5. Given dedifferentiation of cultured mouse hepatocytes and human hepatocytes has been published previously, the authors are encouraged to compare their cells with published mouse hepatocytes head-to-head (CLiP cells, Takeshi Katsuda et al., Cell Stem Cell, 2017) and re-analyzed the published human data to find whether their conclusions stand in human cells.

Response: In accordance with your comment, we cultured adult mouse hepatocytes according to the published protocol (Katsuda et al., *Cell Stem Cell*, 2017) and induced hepatocyte dedifferentiation into CLiPs to compare them with dediHeps. Our data showed that CLiPs required a couple of months to proliferate sufficiently, while dediHeps needed only a couple of weeks. In fact, the proliferative capacity of CLiPs was significantly lower than that of dediHeps (Supplementary Fig. 10a). CLiPs were morphologically identified as epithelial cells and expressed *Alb*, *CK19*, *E-cad*, and *Vim*, similar to dediHeps (Fig. 7a), suggesting that CLiPs also have a hybrid E/M phenotype. Global gene expression analyses revealed that expression levels of *Cdh1*, *Cldn3*, *Ocln*, *Vim*, *Prom1*, *Hnf4a*, and *Hnf1a* were similar, but those of *S100a7a*, *Lamb1*, *Mki67*, and *Krt19* were lower, in CLiPs relative to their expression in dediHeps (Fig. 7b). Meanwhile, several genes encoding

liver enzymes were highly expressed, and the expression level of *Cdx2* was significantly lower, in CLiPs compared to their expression in dediHeps (Fig. 7b). Moreover, PCA revealed that the gene expression signatures of CLiPs were most similar to those of hepatocytes cultured for 1 week (Fig. 7c). Dedifferentiation of human hepatocytes into CLiPs (Katsuda et al., *eLife*, 2019) also activates the expression of the mesenchymal cell markers *VIM* and *LAMB1*, in addition to *PROM1*, *MKI67*, and *KRT19* (Supplementary Fig. 10b). Thus, CLiPs partially resemble dediHeps with a hybrid E/M phenotype but have more characteristics as hepatocytes than dediHeps.

Minor concerns:

1. *The hepatocyte culture system, the maximum number of passages and passages used for liver transplantation should be described.*

Response: In accordance with your comment, we continued the passages of dediHeps in monolayer culture and found that dediHeps could be stably maintained in monolayer culture after more than 20 passages. To examine how long dediHeps can maintain re-differentiation potential, we used dediHeps that have undergone 20 passages in long-term monolayer culture for cell-aggregate formation in 3D cultures. Our data demonstrated that the hepatic maturation of dediHep aggregates could be induced from these dediHeps even after a prolonged period of monolayer culture (Supplementary Fig. 3). In addition, we used aggregates formed from dediHeps with 10 to 12 passages in monolayer culture for transplantation. We described this in the Method section.

2. *Figure 4B: Detailed characterization of dediHep-SOs is required, such as liver hepatic/biliary markers, and the functional characterization of intestinal epithelial cells.*

Response: In accordance with your comment, we conducted co-immunofluorescence staining of Hnf4 α with CK19 or Alb for dediHep-derived SOs. Our data showed that dediHep-derived SOs were formed by cells expressing CK19 and Hnf4 α but not Alb, indicating that these SOs were not biliary organoids that express CK19 but not Hnf4 α and Alb (Supplementary Fig. 6b). Moreover, we examined whether dediHep-derived SOs were composed of functional intestinal epithelial cells. Our data showed that addition of forskolin (FSK) with or without the cystic fibrosis transmembrane conductance regulator

(CFTR) inhibitor CFTRinh-172 to the culture media blocked or induced the swelling of dediHep-derived SOs, respectively (Supplementary Fig. 7). Thus, epithelial cells forming dediHep-derived SOs can functionally respond to FSK in a CFTR-dependent manner and increase the size of the SOs.

3. Figure 4C: the proportion of intestinal epithelial markers in dedihep-SOs? The negative control for the immunostaining is required.

Response: In accordance with your comment, we investigated the proportion of Cdx2-positive cells in dediHep-derived SOs. As shown in Fig. 5c, about 95% of cells that formed dediHep-derived SOs expressed Cdx2. In addition, about 95% and 98% of cells forming these SOs were also positive for Villin and Sox9, respectively (Fig. 5c). These data demonstrate that dediHep-derived SOs are composed of a nearly homogeneous cell population, such as intestinal epithelial cells that express Cdx2, Villin, and Sox9. Moreover, as negative controls, we conducted co-immunofluorescence staining of dediHep-derived SOs using mouse, rabbit, and goat IgG isotype control antibodies and showed the representative micrographs in Supplementary Fig. 6a.

4. Figure 4E: Controls of cdx2-negative cells from the same culture should be included.

Response: As shown in Fig. 5c, dediHep-derived SOs are composed of a nearly homogeneous cell population, such as intestinal epithelial cells expressing Cdx2, Villin, and Sox9. Thus, almost all of cells transplanted into the colon were positive for Cdx2.

5. Figure 5D, 5E: Positive control of intestinal epithelium markers should be included.

Response: In accordance with your comment, we obtained intestinal crypts from 10 week-old adult mice and cultured to form budding organoids (BOs) from intestinal stem cells (ISCs). Then, we conducted co-immunofluorescence staining of E-cad with Muc2, Villin, ChgA, Lyz, or CC3 and Sox9 with EphB2 for ISC-derived BOs, as well as dediHep-derived BOs. Our data showed that characteristics of dediHep-derived BOs closely resemble those of ISC-derived BOs (Fig. 6f). Moreover, we added ISC-derived BOs as samples used for qPCR and showed the data in Fig. 6g and Supplementary Fig. 9.

REVIEWER COMMENTS

Reviewer #1 (Remarks to the Author):

The revised manuscript by Miura and colleagues is much improved. I commend the authors in addressing most of my comments. The characterization of the mesenchymal phenotype over time in culture is much improved. I also appreciate the authors adding controls using freshly isolated hepatocytes in the FAH to show that the dediHep-derived hepatocytes had similar repopulation capacity in vivo. The characterization of dediHep-derived with FIPC-derived SOs is also much improved and addresses my initial concerns. I thank in authors in clarifying the methods section to eliminate any confusion that dediHeps were co-cultured with intestinal cells. I think the addition of comparison analysis of dediHeps with CLiPs significantly improves the paper. The additional studies to clarify the timing and function of Hippo signaling pathway also addresses my initial concerns.

My only remaining comment is regarding the heterogeneity of dediHeps in 3D culture. The authors showed that dediHeps expressed Cdx2 protein at different levels. The new Cdx2 FISH images in figure 4b also suggests that these differences exist at the transcript level as well. I think the authors need additional data in before stating that the dediHeps are a “nearly homogeneous cell population”. The authors state that the micrograph in figure 3a indicates that all dediHeps contribute to cell-aggregate formation, I’m not sure how this conclusion can be made from these images without additional lineage markers.

Reviewer #2 (Remarks to the Author):

In response to this reviewer’s major point #1 (What is the efficiency of spherical organoids and/or colonic tissues formation derived from Cdx2 negative, Cdx2 positive and Cdx2 over-expressed dediHeps?) : Authors increased the concentration of antibody and detected significant more Cdx2-positive cells (revised from 10-15% to 60-90%). Given that most dediHep cells are already Cdx2-positive, why overexpression of Cdx2 is required for the formation of budding intestinal organoids (revised Fig 6A)? Authors should provide evidence/interpretation to these seems contradictable results.

Reviewer #3 (Remarks to the Author):

The authors have comprehensively addressed all of my concerns.

Response to the Reviewers

Reviewer #1:

My only remaining comment is regarding the heterogeneity of dediHeps in 3D culture. The authors showed that dediHeps expressed Cdx2 protein at different levels. The new Cdx2 FISH images in figure 4b also suggests that these differences exist at the transcript level as well. I think the authors need additional data in before stating that the dediHeps are a “nearly homogeneous cell population”. The authors state that the micrograph in figure 3a indicates that all dediHeps contribute to cell-aggregate formation, I’m not sure how this conclusion can be made from these images without additional lineage markers.

Response: Thank you very much again for your kind comment regarding our revised manuscript. As you mentioned, dediHeps constitute a heterogeneous population with respect to the expression of Cdx2. In order to examine whether both Cdx2-positive and Cdx2-negative dediHeps contribute to cell-aggregate formation in 3D culture, we conducted immunofluorescence staining of Cdx2 for dediHep aggregates at 12 hours and 1, 2, 3, 4, and 5 days after initiation of 3D culture. Our data showed that both Cdx2-positive and Cdx2-negative dediHeps contribute to cell-aggregate formation and that Cdx2 expression gradually disappeared from dediHep aggregates in 3D culture (Supplementary Fig. 5a and b). It is noteworthy that the percentage of Cdx2-positive cells in dediHep aggregates at 12 hours after initiation of 3D culture is 57.1% (Supplementary Fig. 5b), which is close to the percentage of Cdx2-positive cells (60.7%) in monolayer cultures of dediHeps (Fig. 4a). These data suggest that both Cdx2-positive and Cdx2-negative dediHeps in monolayer culture contribute equally to cell-aggregate formation under 3D culture conditions.

Reviewer #2:

In response to this reviewer’s major point #1 (What is the efficiency of spherical organoids and/or colonic tissues formation derived from Cdx2 negative, Cdx2 positive and Cdx2 over-expressed dediHeps?) : Authors increased the concentration of antibody and detected significant more Cdx2-positive cells (revised from 10-15% to 60-90%). Given that most dediHep cells are already Cdx2-positive, why overexpression of Cdx2 is required for the formation of budding intestinal organoids (revised Fig 6A)? Authors

should provide evidence/interpretation to these seems contradictable results.

Response: Thank you very much again for your kind comment regarding our revised manuscript. While your concern is valid, obtaining an accurate answer is currently challenging. One possibility is that the regulation of target gene expression by Cdx2 is related to the expression level of Cdx2 itself. It has been reported that, in the adult intestine, Cdx2 regulates genes that are different from those regulated during embryonic development, such as *intestinal alkaline phosphatase*, *fatty acid-binding protein 1 and 2*, *lactase*, *sucrase isomaltase*, *meprin A subunit beta*, and *microsomal triglyceride transfer protein* (Kumar et al., *Development*, 2019). Thus, we analyzed the transcriptome data obtained in this study to examine whether these adult intestine-specific Cdx2 target genes were upregulated in dediHep-derived BOs. Our transcriptome analyses revealed that the expression levels of these genes in dediHep-derived BOs were higher than those in dediHep-derived SOs and FIPC-derived SOs and similar to those in ISC-derived BOs. Thus, it is suggested that the expression level of intrinsic Cdx2 in dediHeps is insufficient to induce the upregulation of adult intestine-specific Cdx2 target genes. We described this in the Discussion section.

Reviewer #3:

The authors have comprehensively addressed all of my concerns.

Response: Thank you very much for accepting our revision.

REVIEWERS' COMMENTS

Reviewer #1 (Remarks to the Author):

The authors have address all of my concerns.

Reviewer #2 (Remarks to the Author):

Althogu an accurate answer to my last question is lacking, the authors have adressed my concern by mentioning this in the discusion section.